# Co-Culture Models: Key Players in In Vitro Neurotoxicity, Neurodegeneration and BBB Modeling Studies

**DOI:** 10.3390/biomedicines12030626

**Published:** 2024-03-12

**Authors:** Ana Rita Monteiro, Daniel José Barbosa, Fernando Remião, Renata Silva

**Affiliations:** 1UCIBIO—Applied Molecular Biosciences Unit, Laboratory of Toxicology, Department of Biological Sciences, Faculty of Pharmacy, Porto University, 4050-313 Porto, Portugal; rita_pmonteiro@hotmail.com (A.R.M.); remiao@ff.up.pt (F.R.); 2Associate Laboratory i4HB—Institute for Health and Bioeconomy, Faculty of Pharmacy, University of Porto, 4050-313 Porto, Portugal; 3UCIBIO—Applied Molecular Biosciences Unit, Translational Toxicology Research Laboratory, University Institute of Health Sciences (1H-TOXRUN, IUCS-CESPU), 4585-116 Gandra, Portugal; daniel.barbosa@iucs.cespu.pt; 4Associate Laboratory i4HB—Institute for Health and Bioeconomy, University Institute of Health Sciences—CESPU, 4585-116 Gandra, Portugal; 5i3S-Instituto de Investigação e Inovação em Saúde, Universidade do Porto, 4200-135 Porto, Portugal

**Keywords:** blood–brain barrier, neurovascular unit, in vitro, co-culture models, neurotoxicity, neurodegeneration

## Abstract

The biological barriers existing in the human body separate the blood circulation from the interstitial fluid in tissues. The blood–brain barrier (BBB) isolates the central nervous system from the bloodstream, presenting a dual role: the protection of the human brain against potentially toxic/harmful substances coming from the blood, while providing nutrients to the brain and removing metabolites. In terms of architectural features, the presence of junctional proteins (that restrict the paracellular transport) and the existence of efflux transporters at the BBB are the two major in vivo characteristics that increase the difficulty in creating an ideal in vitro model for drug permeability studies and neurotoxicity assessments. The purpose of this work is to provide an up-to-date literature review on the current in vitro models used for BBB studies, focusing on the characteristics, advantages, and disadvantages of both primary cultures and immortalized cell lines. An accurate analysis of the more recent and emerging techniques implemented to optimize the in vitro models is also provided, based on the need of recreating as closely as possible the BBB microenvironment. In fact, the acceptance that the BBB phenotype is much more than endothelial cells in a monolayer has led to the shift from single-cell to multicellular models. Thus, in vitro co-culture models have narrowed the gap between recreating as faithfully as possible the human BBB phenotype. This is relevant for permeability and neurotoxicity assays, and for studies related to neurodegenerative diseases. Several studies with these purposes will be also presented and discussed.

## 1. Blood–Brain Barrier: Development and Characterization

The blood–brain barrier (BBB) is a selective and highly specialized structure primarily formed by a tight monolayer of endothelial cells of cerebral capillaries and microvessels. The BBB controls the movement between the bloodstream and the brain interstitial space, protecting the central nervous system (CNS) against pathogens and potentially harmful substances [1,2,3]. Its defense role is due to: (i) the tight paracellular barrier, achieved by the inter-endothelial junctions; (ii) the low levels of transcellular transport; (iii) the high presence of efflux pumps in endothelial cells, and (iv) the presence of metabolic enzymes (acetylcholinesterase, monoamine oxidase, cytochrome P450-related enzymes (CYP450), among others). Regardless of its well-established barrier function, the BBB also allows the inflow of ions, nutrients, and oxygen necessary for the normal neural activity [4,5].

The CNS (brain and spinal cord) is highly vascularized to cover the metabolic demands of this active tissue [5]. In the embryonic phase, brain formation starts with vascularization, which occurs over two different processes: vasculogenesis and angiogenesis. Vasculogenesis consists of the differentiation of mesoderm-derived angioblasts (progenitors of endothelial cells) into blood vessels. This process takes place in the initial stages of the embryonic phase with the formation of the perineural vascular plexus (vascular complex surrounding the neural tube) [5,6]. Angiogenesis is a process described as the formation of new vessels from pre-existing ones, taking place in later stages of the embryonic development and adulthood [5].

In the embryonic stage, after the formation of the perineural vascular plexus, the angiogenesis process sets in, overlapping with the BBB formation (Figure 1A). Being a multistep process, it starts when highly invasive endothelial progenitor cells sprout from the perineural vascular plexus and migrate towards the neuroepithelium of the neural tube [5,6,7,8]. The angiogenesis process is driven by several factors released by the endothelial cells, epithelium, and neural progenitor cells, such as vascular endothelial growth factor (VEGF), fibroblast growth factors, placenta growth factors (PIGF), interleukins, G-protein-coupled receptor (GPR124), and angiopoietin-1, involving the Wnt-β-catenin signaling pathway [1,5,6,8].

After angiogenesis, the barrier maturation continues with the appearance of tight junction proteins, nutrient transporter expression, the increased expression of efflux transporters, decreased levels of transcytosis, and decreased permeability to water-soluble compounds, such as mannitol, potassium, and urea [1,7,8].

The newly formed vessels recruit pericytes, responsible for the stabilization of the new structure membrane [1,8]. Pericytes recruitment is mainly due to the secreted platelet-derived growth factor B (PDGF-B) produced by endothelial cells [6,8].

At the same time, neural progenitor cells originating from the neuroepithelium, using radial glia cells as a guidance structure, migrate towards the cerebral cortex (Figure 1A). During this process, they differentiate into neuroblast and neurons [1,6]. Astrocytes developing from radial glia, like pericytes, limit BBB permeability [8,9]. The development of the BBB phenotype is not inherent to brain endothelial cells. Barrier properties mature when endothelial cells come in close contact with pericytes and astrocytes (Figure 1B) [1,7,8]. Pericytes and astrocytes are two cellular components identified as surrounding elements of the endothelial cells, important not only for embryonic development, but also for the maintenance of the BBB characteristics in adult life [3,7]. Previous studies support the importance of pericytes in the BBB formation, as their absence leads to endothelial hyperplasia and increased BBB permeability [3,10].

Historically, the BBB was first defined as a tight monolayer of endothelial cells that form the walls of vessels/capillaries. Currently the concept of a neurovascular unit is embraced when the BBB is characterized [4,11].

The BBB is an endothelial structure organized in a set of components responsible for maintaining the BBB microenvironment adjusted for its normal functions (Figure 1B). Its typical phenotype and plasticity are obtained by the presence of the neurovascular unit that combines a cellular and molecular complex formed by neurons, astrocytes end-feet, microglia, pericytes, endothelial cells, and extracellular matrix [2,4], individually explored in the following sections.

### 1.1. Endothelial Cells

Endothelial cells represent the principal barrier unit of the BBB. They line the brain microvasculature and are connected by tight junctions and adherent junctions (Figure 2). In close contact with the basal membrane of the endothelial cells are pericytes and astrocyte end-feet (Figure 1B) [11,12]. The brain endothelial cells are different from the endothelial cells of other tissues, especially since they show low levels of transcytosis (the vesicular transport of macromolecules from one side of a cell to the adjacent cells) and a lack of fenestrae (small openings or pores) [3,12].

The tight junctions narrow the paracellular cleft and comprise proteins, such claudin, occludin, and junctional adhesion molecule (JAM) [2,11,13]. Claudin and occludin proteins are connected by homophilic interactions and are also connected to the actin cytoskeleton via zonula occludens proteins (ZO), namely ZO-1, 2 and 3 [2,9,11].

The adherent junctions include platelet endothelial cell adhesion molecules (PECAMs), vascular endothelial cadherin (VE-cadherin), and catenins [2,13]. VE-cadherin is linked to the actin cytoskeleton by α-, β- and γ-catenins [11,13]. Even though they are not described as the main structure limiting the paracellular permeability, adherent proteins are essential for a functional brain barrier [3,14,15]. Furthermore, it has been shown that a prior assembly of adherent junctions supports the formation of tight junctions [16]. Table 1 summarizes the principal features of the proteins that constitute each family of inter-endothelial junctions.

Joining the junctional proteins, the endothelial cells also express several transporters, which help in reinforcing the barrier status [4]. The solute-carrier transporters (SLC transporters) and ATP-binding cassette efflux transporters (ABC transporters) are explored in Section 1.7.

All the specifications previously displayed help to limit the entrance of substances in the CNS. Figure 3 schematizes the main transport mechanisms by which compounds can enter the brain. Substances with low molecular weight, not charged, and liposoluble can cross the BBB by passive diffusion. Small compounds, ions, and small water-soluble solutes can cross it through the narrowed paracellular cleft. The transcellular transport involves specific transporters and receptors. By carrier-mediated transport, small molecules like glucose, amino acids, vitamins, and nucleosides enter the CNS. Large molecules, including transferrin and insulin, cross the BBB by receptor-mediated transport. Lastly, charged plasma proteins cross the BBB by adsorptive-mediated transcytosis [4,5,11,17].

### 1.2. Pericytes

Pericytes are contractile, mesodermal origin cells surrounding the endothelial cells of the blood vessels [4,11]. The presence of smooth muscle actin fibers enables them to control the degree of blood vessel constriction [17].

This cellular component of the neurovascular unit is responsible for vessel stability, regulates capillary diameter, and consequently the blood flow, regulates extracellular matrix protein secretions and levels, and contributes to the BBB integrity and functions [4,8,9].

The contribution of pericytes for the BBB integrity and functional characteristics is associated with their close contact with the endothelial cells. Pericytes enhance the transcellular and paracellular barrier, because they contribute to tight junction formation (occludin and claudin-5) and decrease transcytosis [8,9,17].

### 1.3. Neurons

Neurons are specialized cells responsible for transmitting electrical and chemical signals to other cells. Like the other cells that form the neurovascular unit, they influence the BBB properties and physiology [9,18].

The brain normal activity requires highly metabolic needs and neurons regulate the function of blood vessels to reply to the metabolic requirements [9,19]. Pericytes and astrocytes work alongside with neurons. Both cells seem to communicate with neurons to modulate synaptic strength. They also transmit signals from neurons to the endothelium resulting in vasculature dilation and increase blood flow [9,11].

### 1.4. Astrocytes

Astrocytes are key cellular components for ensuring BBB integrity, supporting the tight junction formation, and modulating the expression of transporters. They release several molecules that enhance and maintain the barrier properties of endothelial cells, such as members of the Hedgehog family (modulate tight junctions), transforming growth factor-β (TGF-β), renin-angiotensin hormone system, cholesterol, and the phospholipid transporter molecule apolipoprotein E (helps to keep brain homeostasis) [4,9,11].

They also support and protect neurons by participating in the clearance and control of neurotransmitters, by providing nutrients (oxygen and glucose) to the neural activity through the regulation of blood flow, and by controlling the number of synapses [4,11]. In addition, they help to regulate immune system [4,8].

Astrocytes communicate with endothelial cells via end-feet projections. This connection mostly helps to regulate the brain water content and potassium levels, once they express aquaporin 4 and ATP-sensitive potassium channel Kir 4.1, respectively [3,8,9].

### 1.5. Microglia

Microglia also influence BBB integrity and constitute the brain central line of defense. The CNS has resident immune cells that, giving their perivascular localization, quickly react to inflammatory triggers [9,17].

Microglia can differentiate into two phenotypes, according to the received stimulus: pro-inflammatory M1 or anti-inflammatory M2. The M1 microglia encourage the BBB opening since they secrete pro-inflammatory cytokines, such as interleukin-1 β (IL-1 β), tumor necrosis factor-α (TNF-α) and nitric oxide (NO). On the other hand, the M2 microglia release transforming growth factor-β and VEGF, promoting immunosuppression and angiogenesis, respectively [17,20]. The M2 microglia is involved in tissue repair and phagocytosis [4].

### 1.6. Extracellular Matrix

The extracellular matrix is composed of proteins secreted from endothelial cells, astrocytes end-feet, and pericytes that contribute to the BBB phenotype and integrity [4,8]. Its components include collagen IV, fibronectin, fibrillins, laminins, glycosaminoglycans, heparan sulfate proteoglycans (HSPG), perlecan, agrin, and regulatory enzymes metalloproteinases [8,9,17].

The matrix metalloproteinases (MMPs) modify the extracellular matrix and, like the other molecular components, regulate the BBB integrity. Studies have shown that pericytes regulate the MMP-9 production and, in their absence, a decrease of that metalloproteinase influences the trans-endothelial electrical resistance (TEER) of endothelial cells [17].

### 1.7. ABC and SLC Transporters

The ABC transporters are efflux pumps expressed primarily in the luminal/apical side (part of the membrane in contact with blood) of the BBB (Figure 2), presenting an important neuroprotective role. They use ATP-hydrolysis (the way in which they obtain the energy needed to transport substances against the concentration gradient) to regulate the transport of substances, thus returning to the circulatory system of unwanted xenobiotics and endobiotics. Indeed, they restrict the entrance of therapeutic agents, contributing to CNS pharmacoresistance [21,22,23,24]. The best-studied subfamilies are ABCB1 (multidrug resistance proteins, MDR1, or P-glycoprotein, P-gp), ABCC (multidrug resistance-associated proteins, MRPs), and ABCG2 (breast cancer resistance protein, BCRP) [3,22,25].

Table 2 assembles the main characteristics of the major efflux pumps expressed in the BBB.

The solute carrier (SLC) group of membrane transport proteins are expressed in the luminal and/or abluminal/basolateral (facing the brain side) membrane of endothelial cells (Figure 2), catalyzing both the uptake and removal of substances from the brain. For this reason, they play a role in controlling the extracellular fluid content, securing the proper functioning of brain activity. Delivery of hydrophilic compounds, which include nutrients indispensable for brain homeostasis, is made via these specialized transporters [3,27].

The SLC families expressed in the BBB involve transporters responsible for carrying glucose, monocarboxylates, creatinine, neurotransmitters/precursors, amino acids, and organic ions, as well as several drugs [3,27]. Organic cation and anion transporters are identified as a primary route for drug delivery to the brain [27].

Table 3 gathers some of the most prevalent SLC families expressed in the BBB, as well as their main functions.

## 2. Cell Cultures as In Vitro BBB Models

Many cell culture models based on primary cells or immortalized brain capillary endothelial cells have been developed over the last years, especially aiming to facilitate the implementation of in vitro studies for developing CNS drugs [29,30].

The regulation and function of human BBBs is far from fully known. It is nevertheless worth noting that there is a great interest in creating cell-based models capable of mimicking the in vivo proprieties of the BBB, namely tight junctions, transporters, enzymes, macromolecules, and immune trafficking [29].

The next chapters will discuss the most commonly used primary and immortalized brain endothelial cell models in monoculture, before introducing the topic of co-cultures. Two non-brain endothelial cells (Caco-2 and MDCK) most used by the pharmaceutical industry, mainly for P-gp expression, are also approached. The models will be accompanied by a description of their main characteristics, including the validation markers that must be present, as well as their main limitations. In addition to the most commonly used endothelial cell models in monoculture, the SH-SY5Y cell line will also be mentioned for its relevance in toxicity assays and in vitro studies performed in neuroscience research. The latest advances in the techniques available for BBB modeling will also be addressed.

### 2.1. Validation Markers for Cell Culture Models

There are important chemical/biological markers, entitled by some authors as validation markers [29,30], that need to be present in a potential BBB model. In fact, tight junctions and the expression of transport systems (characterized in previous chapters) are the two most widely used markers to validate in vitro models of the BBB.

For the cell culture of a BBB model, it is essential that cells are capable of forming a thin monolayer with a large surface area. High junctional tightness, which allows the characterization of paracellular permeability, and the expression of efflux and SLC transporters, which are fundamental for the evaluation of the permeability to endogenous compounds and xenobiotics, represents other aspects. In addition, the uptake capability of macromolecules, especially regarding transcytosis receptor-mediated transport, constitutes another pre-requisite of an in vitro BBB model [29,31]. Table 4 systematizes some of the current validation markers of cell culture models of BBB, the limitations, and their relevance to in vitro studies.

### 2.2. Cell-Based Models’ Origins

The cell-based models developed over the past few years have been limited to the use of primary and immortalized cell lines, obtained from a wide range of species that include mouse, rat, pig, and human [13,31]. There seems to exist some agreement among researchers that the use of primary cells provides a model with greater physiological relevance [13,31,32]. However, each source, regardless of its origin, has disadvantages, especially concerning the yield and barrier properties [13,31]. Table 5 presents the main advantages and disadvantages for each cell model according to source and origin.

The use of primary or low passage cells is favorable because they have higher TEER values and low paracellular permeability. However, the isolation of primary cells is a time-consuming process, and it is more likely to suffer from contamination with non-endothelial cells. Especially animal models demonstrate major disadvantages, namely in the expression of influx and efflux transporters, when compared to the in vivo characteristics of the human BBB [31,35]. To avoid interspecies variability, primary cells of human origin could be utilized, but their use is limited by ethical issues. Alternatively, immortalized human brain capillary endothelial cells can be used. Despite their main limitation (low TEER values), which has driven several studies aiming their optimization, their use is beneficial because they proliferate indefinitely while eliminating the interspecies variability problem [35].

The cells from human and non-human origin used to create in vitro models of the BBB, studied so far, are explored in the next two chapters. For each cell model, we show a description of their general characteristics, including their limitations and the BBB markers they express.

### 2.3. In Vitro Models: Primary Cells

#### 2.3.1. Bovine Brain Microvascular Endothelial Cells (BBMECs)

The BBMECs are a well-established model of the BBB. The cells are isolated from bovine cerebral cortex, by enzymatic digestion, and purified by centrifugation [33,34,35]. They show a spindle-like morphology, forming monolayers with tight junctions, and expressing many transporters and enzymes typical of the BBB [36,37]. Giving the ability to retain many morphological and biochemical properties of the BBB (Table 6), BBMECs have been used in studies of uptake, transport, and metabolism [38,39].

#### 2.3.2. Porcine Brain Microvascular Endothelial Cells (PBMECs)

Like the previous cells, PBMECs are isolated from the grey matter tissue of porcine brain, by enzymatic digestion, and purified by centrifugation [42]. Microscopic examinations demonstrate that the morphology of the cells is consistent with endothelial microvasculature since they adopt a spindle-like, squamous shape [43]. This model has shown high TEER values with a good expression of tight junction proteins, transporters, receptors, and enzymes. Noteworthy, and most importantly, this model has been extensively used since: (i) the isolation of the brain endothelial cells gives a high yield of cells, when compared to that from rat or mouse, therefore less animal sacrifice; (ii) lower interspecies variability (the porcine genome, anatomy and physiopathology reflect the human biology more closely than many laboratory animals. Even disease progression resembles humans, making pigs one of the most suitable animal model to study human disease); (iii) as a by-product of the meat industry, the porcine brain is relatively easy to obtain and unexpensive; (iv) their usage is more ethically acceptable for research; (v) compared to other animals, especially rodent and bovine, BBB models from porcine usually retain many key features of the BBB after isolation and present slower rate of phenotype loss of the BBB in culture; therefore, co-culture with, for example, astrocytes is not essential to induce high TEER values [42,44,45]. The topic of co-culture will be explored later. Table 7 assembles the main BBB characteristics expressed in this model.

#### 2.3.3. Rat Brain Microvascular Endothelial Cells (RBMECs)

The RBMECs obtained from the cortical gray matter of the rat brain have been used in drug uptake and transport studies. Their main features are listed in Table 8. The low number of endothelial cells obtained per animal and the naturally high abundance of pericytes that may physically interrupt the formation of the endothelial cells monolayers represent drawbacks of this model [47,48].

#### 2.3.4. Human Brain Microvascular Endothelial Cells (HBMECs)

The HBMECs present a spindly-like, cobblestone morphology [51,52]. Derived from microvessels isolated from cerebral cortex/temporal tissue from surgical resections of patients with seizures or post-mortem brains, the cells express adherent junctions’ proteins such as β-catenin, enzymes, and other components, as listed in Table 9. Creating models of the BBB using non-human cells is relatively simple, but they are not comparable to humans concerning their characteristics and functions, as well as regarding the notable expression of efflux transporters. However, the use of HBMECs is limited due to the availability of the tissue and the ethical problems regarding the isolation process [51,52,53].

### 2.4. In Vitro Models: Immortalized Cell Lines

#### 2.4.1. BB19

This immortalized cell line is derived from the human brain endothelium and immortalized with *E6* and *E7* genes of the human papillomavirus. It retains some of the endothelial properties, including expression of VE-cadherin, among other characteristics listed in Table 10. Its use as a BBB model is limited due to a high sucrose permeability and very low values of TEER [35,54].

#### 2.4.2. RBE4

The RBE4 is an immortalized rat brain microvessel endothelial cell line with a spindle-like morphology. Although these cells show relative low sucrose permeability, they have been used for drug transport studies. Their main features are listed in Table 11 [34,54,55,56].

#### 2.4.3. b.End3

The b.End3 cell line is an immortalized mouse brain cell line (brain endothelial cells of SV129 mice) transfected with polyomavirus middle T-antigen, presenting a spindle-like squamous morphology. Although widely used for blood vascular research, it has been used for drug uptake and transport studies [57,58]. Even though b.End3 cells functionally express important transporters and proteins of the BBB (Table 12), they only present a moderate barrier property against sucrose diffusion, lacking real discrimination to the permeation of transcellular and paracellular crossing [57,58,59].

Furthermore, previous studies also demonstrated that the b.End3 and b.End5 cell lines express glutathione. A recent study investigated how BBB endothelial cells respond to oxidative stress by exposing b.End3 and b.End5 cell lines to hydrogen peroxide. [59]. A decreasing viability of both cell line following exposure to increasing concentrations of hydrogen peroxide was shown. As hydrogen peroxide concentrations increased, the b.End3 cell line replied with a decline in glutathione content, while the b.End5 demonstrated an initial increase followed by a decrease in the glutathione antioxidant levels [59]. The authors also predict, by making comparisons with other cell lines in terms of cellular glutathione (liver cancer cell line HepG2), that both cell lines seem adequate to modeling the oxidative stress in the BBB and, therefore, to study the glutathione responses to oxidative stress [59].

#### 2.4.4. b.End5

The b.End5 cell line was established from brain endothelial cells of BALB/c mice and immortalized by the infection of primary cells with retrovirus coding for the polyomavirus middle T-antigen. Microscopically, these cells show an oval and spindle-like morphology [37,60]. As mentioned above, a recent study has shown that this immortalized cell line expresses glutathione. In that study, the researchers noted that the b.End5 cell line showed a higher antioxidant capacity when compared to the b.End3 cell line [59]. The b.End5 cells assemble some of the BBB characteristics (Table 13) but fail to establish a tight permeability barrier [37,60].

#### 2.4.5. cEND

The cEND cell line is an immortalized mouse cerebral capillary endothelial cell line that shows a spindle shape morphology. The immortalization was achieved following transformation with polyomavirus middle T-antigen [61,62]. It provides a useful model for studies of differentiation and regulation of the BBB, especially because it retains endothelial features, as VE-cadherin and PECAM-1, tight junctions, and presents high TEER values. This cell line elucidates the cellular response of endothelial cells to different stimuli, including glucocorticoid, estrogen-treatment, and pro-inflammatory mediators [61,62,63,64]. Its main characteristics are gathered in Table 14.

#### 2.4.6. cerebEND

The cerebEND cell line is an immortalized mouse cerebellar capillary endothelial cell line (immortalization with polyomavirus middle T-antigen) with a spindle-like morphology [65]. The cerebellar BBB helps to maintain the homeostasis of the cerebellum and perturbations in its integrity is often connected to many CNS disorders. Like the previously mentioned cell line, it retains endothelial features, as VE-cadherin and PECAM-1, tight junctions, and presents high TEER values [62,65]. The differences between the two cell lines report the higher fragility of the cerebellar barrier to inflammatory stimuli and the higher expression of claudin-3 and 12 in the cerebEND cell line. The higher fragility of this cell line could be explained by the different junctional protein expression, namely lower expression of major tight junction components of the BBB, claudin-1, and occludin. Together with the cEND cell line, it represents an important in vitro model system for the two different brain regions [62,65]. Table 15 lists the main properties of this cell line.

#### 2.4.7. hCMEC/D3

The hCMEC/D3 cell line consists of human temporal lobe microvessel endothelial cells immortalized through the transduction with the catalytic subunit of human telomerase reverse transcriptase (hTERT) and Simian virus 40 large T-antigen (SV40T) [66,67]. This cell line is a remarkably unique in vitro model of the BBB, because it retains most of the morphological and functional characteristics of the human BBB, including VE-cadherin, PECAM-1 and JAM-1 expression, and metabolic enzymes (especially phase I CYP450). These characteristics make this model widely used to analyze the human BBB functions and its modulation by pharmacological agents, xenobiotics, and pathogens, among other stimuli. It has been mostly used in drug transport and uptake experiments, immune migration, and neuroscience studies [35,66,67,68,69].

Despite its advantages, this cell line does not seem to be the most suitable in vitro model for performing permeability studies. Several authors point out the permeability to small molecules (high permeability for sodium fluorescein) and low TEER values, possibly due to an incomplete formation of tights junctions [35,70,71]. Although several studies have shown expression of tight junctional proteins, this cell line expresses claudin-5, occludin, and JAM-1 at low levels [66,69,70]. The main characteristics of this cell line are listed in Table 16.

#### 2.4.8. TY08

The TY08 cell line is an immortalized human brain microvascular endothelial cell line, generated by employing a retrovirus encoding a temperature-sensitive Simian virus 40 large T-antigen gene [72]. With a spindle morphology, this cell line has been characterized in terms of barrier properties, such as the expression of tight junctions, and influx and efflux transporters [72,73] (Table 17). However, the TY08 cells present low levels of functional P-gp, a difficulty to acquire an immortality status, and relatively low TEER values [67,72,73]. In fact, the TY08 cell line has a major disadvantage, corresponding to the immortalization process, which has proven to be inefficient: the cells did not grow beyond the 16th to the 18th passage, even under the permissive temperature conditions.

#### 2.4.9. TY09

The TY09 model was acquired introducing in the TY08 cell line the catalytic subunit of human telomerase to generate a useful cell line capable of growing over 50 passages, while retaining the properties of the BBB, like TY08 cells. TY09 is used in drug permeability studies and in studies aiming the elucidation of the mechanisms underlying many neurological diseases [73,74,75]. Table 18 presents the main characteristics of this cell line.

#### 2.4.10. TY10

The TY10 cell line is a new generation of conditionally human immortalized cells derived from TY08 cells [35,76]. The immortalization process was carried out by transfection with a plasmid expressing temperature-sensitive Simian virus 40 large-T antigen and the catalytic subunit of human telomerase [77,78]. They assemble characteristics of TY08 and TY09 cells, express VE-cadherin, and present a high expression of tight junction proteins, in particular claudin-5, occludin, and ZO-1 [35,78,79]. Table 19 shows its main characteristics.

#### 2.4.11. HBEC-5i

The HBEC-5i cell line is a human cerebral cortex microvascular endothelial cell line immortalized after transfection with a plasmid coding for Simian virus 40 large T-antigen [80,81]. With a cobblestone-like morphology, these cells are used in studies related to cancer, endothelium function, cell trafficking, and malaria host–parasite interactions [80,81,82]. Retaining many of the characteristics of the brain endothelial cells (Table 20), this model exhibits advantages when compared to other models, like the hCMEC/D3 cells, including the ability to grow in a monolayer on inserts without adjuvants (previous studies suggested that the use of adjuvants have an impact on the expression of ABC transporters). Therefore, these cells are easier to use routinely in a laboratory to screen drug interactions, for example. Additionally, these cells are also able to generate high TEER values and have shown very low permeability to dextran [81].

#### 2.4.12. HBMEC/cibeta (ciβ)

The HBMEC/ciβ cell line is a clonal human brain microvascular endothelial cell line/conditionally immortalized clone β obtained after the immortalization process carried out by the introduction of temperature-sensitive Simian virus 40 large T-antigen and a human telomerase reverse transcriptase [66,83].

With a spindle-like morphology, it forms a monolayer at the confluence, and exhibits essential markers of the BBB, including PECAM-1, VE-cadherin, β-catenin, functional efflux transporters, and barrier properties that restrict the intracellular crossing of small molecules (studies made essentially with sucrose and sodium-fluorescein) [66,83,84]. Table 21 summarizes the main characteristics of the HBMEC/ciβ cell line.

#### 2.4.13. HBMEC/ci18

The HBMEC/ci18 cell line is a clone of a conditionally immortalized human brain microvascular endothelial cell line recently obtained by introducing the two most commonly used immortalized genes: human telomerase reverse transcriptase and the temperature-sensitive Simian virus 40 large T-antigen [85]. It shows elongated spindle-shaped cell morphology and possesses BBB characteristics, such as the expression of VE-cadherin, β-catenin. Although similar to HBMEC/ciβ cells, because they are derived from the same parental conditionally immortalized mixture cells (parental mixture cells were individualized using a serial dilution method and the clone 18 was identified), the HBMEC/ci18 cell line appears to be functionally more appropriated [85,86]. The main features of the HBMEC/ci18 cell line are listed in Table 22.

#### 2.4.14. SH-SY5Y

The neuroblastoma SH-SY5Y cell line, a subclone of the human neuroblastoma cell line SK-N-SH derived from a metastatic bone tumor, expresses moderate levels of dopaminergic markers and possesses biochemical and functional properties of neurons. It has been used in studies requiring neuronal-like cells for modeling neurodegenerative diseases and for exploring mechanistic hypothesis in neuroscience studies [87,88]. It is a popular model for Parkinson’s disease (PD) research, since it expresses many characteristics of dopaminergic neurons, like tyrosine hydroxylase, dopamine-β-hydroxylase, and the dopamine transporter (Table 23) [89,90].

This cell line can acquire a functionally more mature neuronal phenotype following differentiation with several agents, including retinoic acid and phorbol esters [88,89,90]. Neuronal differentiation renders cells morphologically more similar to primary neurons, comprising a set of events that includes the formation and extension of neurotic processes, an increase of the electrical excitability of the plasma membrane, the formation of synaptophysin-positive functional synapses and the induction of neuron-specific enzymes, neurotransmitters and neurotransmitter receptors [88,89].

Depending on the differentiation-inducing agents, their differentiation can yield different neuronal phenotypes, particularly cholinergic, adrenergic, or dopaminergic, whose characteristics are shown in the Table 23 [88,89].

#### 2.4.15. Non-Brain Endothelial Cell Lines

##### Caco-2

The Caco-2 cell line is a human intestinal epithelial cell line, derived from a colon adenocarcinoma, used in human drug absorption studies. It forms a tight paracellular barrier and has a high expression of P-gp, two factors that make this cell line, when compared with others BBB models, a good model to study the BBB permeability (mostly passive diffusion studies and ligands for efflux transporters) [55].

The characteristics of Caco-2 cell line are present in Table 24.

##### Madin–Darby Canine Kidney (MDCK)

The Madin–Darby Canine Kidney (MDCK) cell line is an immortalized cell line widely used by the pharmaceutical industry in drug penetration studies, along with MDCK-MDR1, a subclone over-expressing the human MDR1 transporter [55,91]. It forms a tight paracellular barrier and allows studying protein trafficking, polarity, and junctions in epithelial cells [91]. Regarding the studies of passive diffusion of compounds, MDCK cells show a weaker correlation when compared to other BBB models or the Caco-2 model. However, its subclone MDCK-MDR1 allows a more accurate prediction regarding the ligands of the efflux transporters [55,91].

Despite cytoarchitectural differences, just like the Caco-2 model, the MDCK cells gather characteristics of the BBB, which are displayed in Table 25, including P-gp overexpression and enzymatic activity [55,91].

### 2.5. Future Perspectives: Microfluidic and Stem Cells

In vitro BBB models have been developed over the years using cultured brain microvascular endothelial cells (BMECs) obtained mainly from non-human sources. Based on several studies that have highlighted differences in the properties of BBB models obtained from different species, especially with regard to the brain pharmacokinetics, claudins, and the expression of ABC/SLC transporters, it has become more evident that the most reasonable solution would be to create a human-based model, what might allow for more accurate studies of BBB function [92,93].

In vitro BBB models from human primary BMECs have drawbacks that include limited cell availability and loss of phenotype during cell culture (from batch to batch). The use of immortalized human cells has been established by the transduction of tumor genes to try to address the limitations of the primary sources, as already explored. However, those cell lines have also failed as ideal models, particularly showing general lower barrier functions when compared to primary cells [92,93]. Nonetheless, BBB development, maintenance and disease states are difficult and time-consuming to study in vivo. In vivo studies are limited to non-invasive methods, such as magnetic resonance imaging, causing researchers to constantly try to find other in vitro models as an alternative for simpler analyses. Additionally, the BBB endothelium is very difficult to harvest from embryonic animal or human brain, greatly limiting the study of the BBB development [1]. Recently, new advances in the field of stem cells have opened pathways for the development of human BBB models that circumvent these limitations. Stem cells are a very promising cell source for the development of in vitro models, given their ability to expand by self-renewal and to differentiate into more specialized cells, particularly into brain endothelial cells [1,92,94].

Human stem cells, such as pluripotent or multipotent cells (like hematopoietic progenitor, neural progenitor cells), can be isolated during development and from adult tissues, and their inherent properties are consequently dependent on the stage and site from which they are isolated. Table 26 systematizes the different types of stem cells and their main characteristics [1,92].

Human-induced pluripotent stem cells (iPSCs) present a unique opportunity for generating HBMECs forming the BBB, as they show limitless proliferation and are able to differentiate into nearly any cell type [95]. Furthermore, iPSCs can also be used to investigate the phenotype of several genetic diseases. This is achieved by isolating cells from a patient with the genetic disease under study, establishing an iPSC line, and differentiating that line into the cell type(s) affected by the disease [1]. However, although iPSCs are well-suited for generating substantial quantities of brain-like endothelial cells and for studying BBB development and disease in vitro [1,92], regrettably, the time, cost, and specialized knowledge needed to differentiate iPSCs into purified BMECs obviates their widespread application [95]. Some of the main characteristics of iPSCs are depicted in Table 26.

To overcome some of the limitations of using iPSCs, namely the time needed for cell differentiation, Hollman and colleagues, in 2017, proposed an accelerated differentiation of human iPSCs to BBB endothelial cells, by using a defined medium, termed E6 [Dulbecco’s modified Eagle’s medium (DMEM)/F-12 containing ascorbic acid, sodium bicarbonate, selenium, human transferrin, and human insulin)]. This expedites the differentiation of iPSCs into BMECs, while achieving a performance comparable to the BMECs generated using well-established methods [95]. Outstandingly, by employing E6 medium in conjunction with updated culture techniques, the differentiation period of iPSCs into BMECs was successfully shortened from thirteen to eight days. By immunocytochemistry, it was confirmed that, similarly to unconditioned medium (UM)-derived BMECs, E6-derived BMECs presented a robust expression of several endothelial markers and their signature tight junction proteins (BBB markers), including claudin-5, occludin, GLUT-1, PECAM-1, and VE-cadherin, demonstrating the acquisition of a BBB phenotype. Furthermore, E6-derived BMECs consistently demonstrated TEER values exceeding 2500 Ω × cm^2^ across multiple iPSC lines (with the highest recorded TEER value of 4678 ± 49 Ω × cm^2^). Moreover, the passive barrier properties observed by the obtained TEER values were further confirmed by the paracellular permeability of fluorescein, which was smaller than 1.95 × 10^−7^ cm/s for all lines. The activity of efflux transporters, such as P-glycoprotein and MRP family members, was also evaluated and no significant differences were detected between E6-derived BMECs and UM-derived BMECs, highlighting that the modification of the differentiation medium did not impact the activity of efflux transporters expressed on BMECs. These findings clearly suggest that the use of E6 medium for the differentiation of iPSCs into BMECs, along with updated seeding techniques, shortens the differentiation period and produces BMECs with comparable functionality to conventional UM-based differentiation protocols. Furthermore, knowing that iPSCs-derived BMECs can acquire an increased barrier phenotype when co-cultured with astrocytes and pericytes, a co-culture model was then developed by co-culturing iPSCs-derived BMECs with iPSC-derived astrocytes and primary human brain pericytes. The impact of co-culturing iPSC-derived BMECs with astrocytes and pericytes was evaluated by the measurement of TEER values. BMECs differentiated using the E6 medium were responsive to the inductive signals from astrocytes and pericytes, achieving a maximum TEER value of 6635 ± 315 Ω × cm^2^. Overall, and given that the robustness and reliability of the developed differentiation protocol was clearly validated across distinct iPSC lines, obtaining BMECs through an accelerated differentiation protocol, with comparable performance and with reduced production costs, this method could be successfully applied in the development of more accessible iPSC-derived in vitro BBB models for a diverse range of applications [95]. Several other studies reporting the use of iPSC-derived BMECs co-cultured with other cells of the neurovascular unit, such as neurons, astrocytes, pericytes and microglia, will be explored in the following sections.

**Table 26 biomedicines-12-00626-t026:** Stem cell types relevant for modeling the BBB and their characteristics.

Types of Stem Cells	Characteristics	References
**Embryonic stem cells (ESCs)**	- Obtained from the inner mass of a blastocyst-stage embryos; - Named pluripotent, meaning they can give rise to every cell type in the human body in an unlimited quantity; - Provide a renewable resource for studying normal development and disease, and potential therapies.	[94,96]
**Neural stem cells (NSCs)**	- Named multipotent, having a decreased potential of self-renewal and differentiate within a lineage under defined conditions; - Isolated from the embryonic CNS; - Can be differentiated into neurons, astrocytes, and oligodendrocytes.	[1,94]
**Mesenchymal stem cells (MSCs)**	- Are multipotent stem cells, which may be present in the umbilical cord, bone marrow, placenta, and fat tissue; - Easily isolated, with several studies showing their enormous potential in distinct research areas, such as cell tissue engineering, gene therapy and cellular biology; - Potential for applications in the treatment of brain diseases; - Possess anti-inflammatory and immune-modulatory properties, causing a reduction in inflammation on injured tissue; - Studies highlighted the ability of this type of stem cells to secrete growth factors that facilitate the regrowth of neurons; - Several studies showed a strong phenotypic similarity between these cells and pericytes.	[94,97]
**Induced pluripotent stem cells (iPSCs)**	- Obtained by converting somatic cells/tissue-specific stem cells to a pluripotent state; - Critical tools to learn more about normal human BBB development, and disease onset and progression; - Can be differentiated into several types of cells, such as cardiomyocytes, β-pancreatic cells, neurons, and glia; - Enable to study the phenotype of several genetic diseases, Alzheimer’s disease, familial dysautonomia and Rett syndrome, by isolating cells from people who manifest the disorder, creating an iPSC line and differentiating it to the cell type most affected by the disease; - Utility for screening prospective new drugs and for assessing drug delivery of the potential therapeutic agents.	[1,92]

Current co-culture models of the BBB are developed mainly using the Transwell culture systems, one of the most used platforms to study barrier formation, cell migration, and drug delivery [92,98]. The Transwell cell culture inserts (Figure 4) are sterilized, convenient, easy-to-use permeable supports, designed to recreate a cellular environment as close as possible to the in vivo state. However, they fail to reinvent aspects related to the three-dimensional (3D) structure of the BBB, as well as the influence of blood flow [92,94]. The use of microfluidic devices has grown in the last decade, and they basically consist of small devices made of small chambers and micro-scale fluidic circuits [99]. Microfluidic allows the mimicry of the in vivo 3D architecture, creating a more realistic microenvironment, while requiring less cell number and reducing the reagents needed; allows to take in consideration the effect of shear stress (which allows the differentiation of the endothelium into a BBB phenotype); enables the real-time monitoring of permeability, cell extravasation, and the dynamics of TEER measures; and facilitates the use of cellular diversity to obtain a more trustworthy human BBB model. Yet, these models have some limitations, especially the fact that the manufacturing process is complex; there is a lack of standardization of device setup parameters, and requires specialized personnel, since it is a recent technology [13,92,94,99]. This topic will be explored in more detail in Section 3.2.

## 3. Co-Culture Models

This section will focus the evolution of the co-culture models over the past years, with the advantages and limitations of using co-culture models in neurosciences, giving particular attention to their main fields of application: BBB permeability evaluation, neurotoxicity assessments and research on neurodegenerative diseases (elucidation of mechanisms underlying the pathophysiology of the disease and the discovery of potential new therapeutics).

### 3.1. Advantages and Limitations

Low TEER values and low expression of important efflux transporters are the two limitations most often pointed out to the monoculture models developed over the years to study the BBB [1,94]. One of the emerging concerns of the researchers was related with the improvement of the monoculture models of endothelial cells, since brain endothelial cells lose their characteristics when in culture. Moreover, the remarkable importance of the neurovascular unit cells in inducing and maintaining the properties of the BBB was established and recognized as fundamental. As a result, monoculture models were gradually replaced by co-culture models [1,94,100].

The shift to multicellular models, in which one or more cells of the neurovascular unit are introduced, has greatly contributed for improving the potential of the models, especially in the case of drug permeability screening. The most widely mentioned co-culture models are those created using astrocytes and/or pericytes together with endothelial cells, as they play a very important role in developing a tight paracellular barrier and in modulating brain endothelial cells’ functions [3,94]. The models can be either syngeneic or developed using cellular components of different origins (e.g., bovine brain endothelial cells in co-culture with rat astrocytes). The advantages of this interaction between different cellular components include an increased expression of transporters and the induction of tight junctions and cell polarity in brain endothelial cells [32,100].

Table 27 compiles the main advantages of using the cellular components that form the neurovascular unit in co-culture with brain endothelial cells.

Thus, the transition from monoculture to co-culture models has narrowed the gap that existed in trying to recreate, as faithfully as possible through in vitro models, the actual microenvironment of the human BBB. Despite the benefits, some shortcomings are attributed to this enhancement. As a result of the synergistic combination of various cell types, the limitations regarding the permeability and TEER values have been effectively improved, but, even then, the models fail to attempt to approximate these values to those associated with the human BBB phenotype. The interspecies variability stands out, and the issue of obtaining neurovascular unit cells of human origin also becomes an obstacle [1,32,94].

The execution of co-culture protocols offers flexibility in the experimental design, even allowing mechanistic studies on the contribution of each cell type in toxicity studies, for example. It should also be noted that working with different culture systems increases the complexity of the work, as each one requires different treatments at different times. It becomes fundamental (i) to know the morphology of the cells to ensure their viability throughout the assay; (ii) to know the correct time to change the medium, so as not to cause cell death and destabilize the growth of other cells; (iii) to know the correct timing of seeding (the model may not work as efficiently as anticipated and the TEER values obtained may not match the predicted values); and (iv) to guarantee the absence of contaminations. This knowledge ensures the reproducibility of the results [51,103].

### 3.2. From Static to Dynamic In Vitro Models of the Blood–Brain Barrier

In addition to the impetus in the structural shift from monoculture to co-culture models, the complexity of the different forms of modeling, in vitro, of the BBB has also evolved. Early BBB studies relied on brain endothelial cells in monoculture in Petri dishes. Using this simple and inexpensive method, cells were used for biochemical, physiological, and cytotoxicity studies. As a main advantage, the use of the Petri dish allows the use of a large number of cells and the easy identification of them. However, as a key limitation, this model does not allow the performance of drug permeability assays [94,104]. Therefore, and in order to eliminate its main limitation, this method was replaced by another well-known, and previously mentioned, static method. In Transwell systems (Figure 4), cells are seeded and submerged in the appropriate medium, allowing to replicate the “blood side” and the “brain side”. This method also allows the co-culturing of brain endothelial cells with other types of cells from the neurovascular unit, and also allows the detaching of the cells from the platform and the collection of them to carry out further studies [26,94]. In these models, the different types of cells can be placed in direct or indirect contact with each other (Figure 4). This fact can also influence the phenotype of the BBB obtained [26]. In a study conducted by Malina and colleagues, in 2009, three different configurations of in vitro BBB were described and compared: (i) primary culture of porcine brain endothelial cells in monoculture; (ii) a non-contact co-culture model of porcine brain endothelial cells with rat brain glial cells, and (iii) a contact co-culture model of porcine brain endothelial cells with rat brain glial cells. They found that the contact co-culture model generates better results concerning the BBB tightness and P-gp functionality and, consequently, gives rise to an improved and reliable model with similar characteristics to those described in vivo [102].

Although widely used, the shortcomings previously mentioned to the Transwell systems can contribute to change the ability of models to make the most correct and reliable in vitro predictions of expected responses in humans [26,94].

The need for studying the contribution, not only of other cells such as astrocytes and pericytes, but also of the blood flow has driven the creation and development of dynamic models [104]. One of those models is the dynamic in vitro (DIV) system. This model uses co-culture and consists in creating intraluminal flow through hallow fibers, recreating artificial capillary-like supports. The brain endothelial cells are cultured inside of a closed chamber, located in the lumen of the hollow fibers. Those cells are then exposed to flow, while other cells are seeded in the extraluminal compartment. The intraluminal flow is regulated by a pulsatile pump, which can be controlled to generate the desirable pressure [94,104]. Recently, other dynamic models using 3D systems have been developed. The latest advances in the previously mentioned microfluidic have allowed the creation of more complex and reliable models, popular in research fields, called “organ-on-a-chip”, which will be detailed below [104,105].

The more conventional types of 3D cell cultures include the hanging-drop method, the forced-floating method, matrices and scaffolds, and agitation-based approaches [106].

The hanging-drop method consists of creating spheroids, using a simple plate. A small number of cells are seeded, and then the plate is inverted, causing the seeded cells to form a hanging droplet. Hence, the cells are concentrated at the tip of the drop, remaining in place due to surface tension. Over time, the spheroids form compact structures with a more homogeneous structure [106].

In the forced-floating method, by using a simple 96-well plate, the cells are grown to form spheroids in a way that prevents them from attaching to the surface, thereby promoting cell-to-cell interactions. The typically used substance to prevent the adhesion of cells to the surface is the poly-hydroxyethylmethacrylate (poly-HEMA), but also agarose is described for coating [106].

The matrices’ and scaffolds’ technology resorts to an extracellular matrix to produce 3D-spheroids. There is sterile extracellular matrix commercially available, being used to grow cells within or on top of the gel. For scaffolds-based culture, the most commonly used substances to construct prefabricated systems are collagen, laminin, and alginate. Those scaffolds models include a group of fibers, whereby the cells can easily migrate and attach near to the other cells. The growth of the previously seeded cells allows the interstitial space between the fibers to be filled, thus giving rise to a 3D morphology [106].

The agitation-based approaches allow the cells to grow without adhering to the walls, consequently forming a stronger cell-to-cell interaction once the cells are placed in suspension in a container and a device keeps them in constant motion. The two most widely used technologies are the spinner flask bioreactor and rotating cell culture bioreactors. In the first, the material to be used includes a container and a stirring element. The cell suspension is continuously stirring and, varying the magnitude of the container that holds them, the spheroids can have different sizes. In the rotating cell culture bioreactors, the only difference from the preceding one is that the entire container is in constant rotation. There are other technologies involving bioreactors, although not as widely used as the rotatory perfusion systems (allow a continuous supply of medium to the cell chamber from external media bottles) and compression bioreactors (provides, in vitro, both mechanical and physiological stimuli for the cells to be more similar to the in vivo conditions; often used in cartilage engineering) [106].

The combination of 3D models of cell cultures with microfluidic technology has made it possible to obtain systems that are more compatible with the BBB characteristics in humans. Therefore, another type of dynamic model is the “Lab-on-a-chip” or micro total analysis systems, which are microfabricated, mostly by photolithography techniques [105,106]. The different platforms used to create these models are glass- or silicon-based systems (glass offers the advantage in high resolution microscopy and provide a stable surface in terms of reproducible and reliable electroosmotic flow); polymer-based systems; paper-based 3D culture platforms (chromatographic papers are used to design hydrophobic barriers); gel-based and gel-free matrices (hydrogels and derivates, as collagen, matrigel, chitosan, and alginate, are used to support, in an in vivo extracellular matrix the mechanical and biochemical behavior of cells) [105,106].

The applications of the microfluid technology “organ-on-a-chip” (that simulates mechanics functions and physiological responses of an entire organ, in this case the brain) encompass: (i) a better analysis of the pharmacokinetic profiles of drugs, by co-culturing different cells; (ii) studies of cell biology (differentiation, neurite extension of neuronal cells and cell migration profile); (iii) modeling and study of neurodegenerative diseases, as Parkinson’s and Alzheimer’s diseases; (iv) to perform cell uptake and cytotoxicity assays of nanodrugs or other candidate compounds that target the CNS; (v) brain tumor research; (vi) identification of neurotoxic compounds. The “brain-on-a-chip” is also an improved model for understanding the link between inflammatory stimuli and neurodegeneration [106,107,108,109].

The following table collects the main advantages and disadvantages of the dynamic BBB models described over this topic (Table 28).

### 3.3. A Compendium of the Applicability of Co-Culture Models in Permeability Studies

This sub-section compiles some in vitro studies reported in the literature using different co-culture models for the evaluation of the permeability across the BBB. The tables organize the studies into categories (studies of primary endothelial cells, immortalized endothelial cells, and stem cells), highlighting the advantages of including other cell types, previously explored, with endothelial cells, as well as some limitations encountered by the authors in validating the models.

As previously mentioned in Section 2, primary cells, despite the time-consuming and laborious isolation process, have advantages related to the high TEER values and to the more relevant physiological characteristics, when compared to immortalized cell lines [31,33]. The intention of co-culturing the primary cells with, for example, astrocytes aimed to create improved models (given the discovery of the importance of the communication between the various cell types of the BBB for its phenotype), that truly recapitulate the transport and permeability mechanisms of BBB [110]. The following table gathers the main advantages of co-culturing primary endothelial cells with other cell types (Table 29).

The immortalized cell lines were obtained to overcome the limitations of primary cells, regarding the difficulty of obtaining cells of human origin, to decrease the interspecies variability and their limited lifespan [35]. One of the most widely used immortalized human cell line in in vitro studies is the hCMEC/D3 cell line, although the tightness of the monolayer is low [26]. A study conducted by Hinkel and colleagues, in 2019, performed a parametric investigation in both static and dynamic cell culture models, aiming to improve the hCMEC/D3 cell line limitation. The study evaluated several points, namely the medium supplementation, contact and non-contact co-cultivation with the human astrocytic cell line SVGmm and the implementation of flow conditions that, as formerly mentioned, are beneficial. The dynamic model used in the study was a dynamic micro tissue engineering system, which allows the consideration of the vascular microenvironment and constantly fills the compartments with fresh medium. Changes in TEER values were monitored. The authors concluded that, regarding the TEER values, hCMEC/D3 cells did not respond positively to any tested parameter [70]. This observation was in support with other studies [35,70,71]. The hCMEC/D3 cells are widely used in in vitro assays, but these limitations support that further improvements or alternative cell cultures are needed to obtain more accurate and reproducible results.

Table 30 collects the advantages and limitations of co-culturing endothelial cell lines with other cell types.

The iPSCs are a more recent type of cells used in in vitro assays. These cells are isolated and, after differentiation into brain endothelial cells, present some of the key BBB characteristics, namely the expression of tight junctions and functional ABC transporters [26,92].

Despite displaying high values of TEER when in monoculture, studies show that the tightness of the monolayer benefits of the co-culture with other types of cells, such as rat astrocytes [26]. Table 31 compiles the advantages of co-culturing stem cells with astrocytes, pericytes, and neurons.

In 2017, Wang and colleagues developed a microfluidic BBB model to mimic the in vivo-like barrier properties for drug permeability screening. For that purpose, human iPSCs-derived BMECs were co-cultured with rat primary astrocytes on the two sides of a porous membrane on a pumpless microfluidic platform for up to 10 days. The developed BBB-on-a-chip model demonstrated a significant barrier integrity, as evidenced by continuous tight junction formation and in vivo-like values of TEER. Indeed, the TEER levels peaked above 4000 Ω·cm^2^ on day 3 on the chip and were sustained above 2000 Ω·cm^2^ for up to 10 days (demonstrating highly sustained TEER values). The suitability of the developed microfluidic BBB model to be applied in drug permeability studies was also assessed by using model drugs, such as cimetidine, caffeine, and doxorubicin, as well as large molecules (FITC-dextrans). The obtained permeability coefficients were similar to those obtained in in vivo studies. Overall, the developed BBB-on-a-chip model closely mimics physiological BBB functions for a prolonged period and may represent a valuable tool for the in vitro screening of drug permeability [124].

Delsing and colleagues developed, in 2018, a human iPSC-derived model of the BBB that included iPSC-derived endothelial cells in co-culture with iPSC-derived pericytes, astrocytes and neurons. Such endothelial cells present in the co-culture model exhibited high TEER values (significantly increased in the co-culture compared to the monoculture for iPSC-derived endothelial cells), functional P-glycoprotein-mediated efflux (evaluated by assessing rhodamine 123 permeability), and the ability to discriminate between CNS permeable and non-permeable substances. Additionally, an upregulation of tight junction proteins (e.g., claudins) and neurotransmitter transporters, as well as changes in several signaling pathways (e.g., WNT, TNF, and PI3K-Akt) were also observed upon co-culture. Overall, these findings suggest that the co-culture of iPSC-derived endothelial cells with other cell types affects the maturity of the endothelial cells, promoting barrier formation at both the functional and transcriptional levels [125].

In 2019, a human in vitro BBB model using iPSC-derived endothelial cells was developed by Li and colleagues to be successfully applied in the in vivo prediction of drug permeability. The developed model exhibited several BBB characteristics, namely the presence of distinct tight junction proteins (claudin-5, ZO-1, and occludin) and endothelial markers (von Willebrand factor and Ulex). Moreover, the model showed high TEER values (1560 Ω·cm^2^ ± 230 Ω·cm^2^), as well as γ-glutamyl transpeptidase activity. Remarkably, by co-culturing iPSC-derived endothelial cells with primary rat astrocytes, a significant increase in the TEER values was observed (2970 Ω·cm^2^ to 4185 Ω·cm^2^). Furthermore, the upregulation of key BBB-related genes was confirmed in human iPSC-derived endothelial cells when compared to primary human BMECs, including BCRP and P-gp. Noteworthy, drug transport assays were performed for nine CNS compounds. A strong correlation was observed between the data obtained for the permeability of non-P-gp/BCRP and P-gp/BCRP substrates across the developed model and the data obtained in rodents following in situ brain perfusion of the same compounds. Overall, these findings suggest that the developed model can be successfully applied for the screening of CNS compounds, in prediction of the in vivo BBB permeability of several compounds or even to study the BBB function/biology [126].

Given the complexity of the neurovascular unit, Canfield and colleagues hypothesized that the BBB phenotype of iPSC-derived BMECs could be improved upon co-culture with iPSC-derived neurons and astrocytes, thus developing a robust multicellular human BBB model. Initially, iPSC-derived neurons and astrocytes were cultured with primary rat BMECs, and an increased barrier tightening was observed. Co-culture of iPSC-derived neurons and astrocytes with iPSC-derived BMECs resulted in a significant increase in TEER value, decreased passive permeability, and the enhancement of tight junction continuity in BMECs, while P-gp efflux activity remained unchanged. A neural cell mixture composed of one neuron to three astrocytes resulted in the most optimal induction of BMEC properties. Lastly, by using BMECs, astrocytes, and neurons derived from the same donor iPSC source, an isogenic multicellular BBB model was successfully achieved [127]. Later, in 2019, the same authors developed a co-culture model of iPSC-derived BMECs, neurons, astrocytes, and brain-like pericytes to better recreate in vitro the complexity of the neurovascular unit. For that purpose, BMECs were co-cultured with neurons, astrocytes, and/or pericytes, alone or in several combinations, and the barrier properties were then evaluated. The most significant barrier tightening was observed when BMECs were co-cultured with pericytes followed by a mixture of neurons and astrocytes (1:3), as demonstrated by a significant increase in the junctional localization of occludin. Moreover, under baseline BMEC monoculture conditions, BMECs also expressed active P-gp, and this expression was maintained regardless of co-culture conditions. Lastly, co-culturing of BMECs with brain-like pericytes significantly decreased the rate of non-specific transcytosis across BMECs. Overall, the developed co-culture model of the neurovascular unit, with different cell types originated from the same donor, allowed the development of an isogenic model that could represent a useful tool to model the human neurovascular unit in health and disease, including to better understand the interplay between the different cell types [128].

In 2021, Noorani and colleagues developed and validated a quasi-physiological microfluidic BBB model suitable for brain permeability studies. By juxtaposing iPSC-derived BMECs with primary human pericytes and astrocytes, a co-culture model was developed, which exhibited BBB-specific characteristics, including low paracellular permeability, efflux activity, and osmotic responses. Indeed, the obtained BBB model exhibited continuous tight-junction patterns, low permeability to mannitol and sucrose, and quasi-physiological responses to both hyperosmolar opening (observed through a decreased BBB integrity) and P-glycoprotein inhibition (observed through an increased permeability of rhodamine 123, a P-glycoprotein substrate). Furthermore, the creation of a tight barrier and the extended long-term viability of the developed model, which is suitable for time-course studies, was ensured by the environmental signals provided by the astrocytes and pericytes located at the abluminal side of the vascular channel. Overall, this innovative co-culture microfluidic platform demonstrated the capability to replicate a quasi-physiological brain microvasculature, thereby facilitating the creation of a highly predictive and translationally relevant BBB model [129].

**Table 31 biomedicines-12-00626-t031:** Advantages of co-culture of stem cells with different types of cells.

Stem Cells	Co-Cultivated with	Advantages/Observations	References
**iPSCs-derived brain endothelial cells**	- Human astrocytes	- Up-regulation of functional GLUT-1 and P-gp; - Formed a tight barrier with consequent high TEER values and low permeability coefficients; - Expression of BBB receptor-mediated transcytosis transporter.	[92,93,130,131,132]
	- Human pericytes	- There was no benefit neither in TEER values, nor in the permeability coefficients.	
	- iPSC-derived astrocytes - iPSC-derived neurons and astrocytes	- Significant increase in TEER values (with maximum values approximately 9000 Ω·cm^2^); - Reduced passive permeability; - Improved tight junction localization when compared to the monoculture; - P-gp expression and activity remain unchanged.	
	- Rat astrocytic C6 glioma cell line	- Induced differentiation of iPSC into endothelial cells; - Increased TEER values; - Decreased dextran permeability; - Up-regulation of the tight junction proteins claudin-5, occludin, and ZO-1; - Up-regulation of BCRP, P-gp, MRP4, and GLUT-1.	

iPSCs—induced pluripotent stem cells; GLUT—glucose transporter; P-gp—P-glycoprotein; TEER—transendothelial electrical resistance; ZO—zonula occludens; BCRP—breast cancer resistance protein; MRP—multidrug resistance-associated protein.

### 3.4. Relevance of In Vitro Co-Culture Models in Neurotoxicity Assessments

Humans are exposed daily to several potentially toxic chemicals present in the environment that, given their lipophilic nature, can easily cross the BBB and induce neurotoxicity. Among these environmental chemicals are the heavy metals, such mercury, and organic chemicals, such as pesticides and industrial surfactants [133,134]. In fact, several studies linked this exposure to neurodevelopment disorders and neurodegenerative diseases [133,134]. Given the inherent limitations of primary and immortalized endothelial cells, and that even with co-culture they still fail to approach the TEER values observed in vivo, an easy solution would be to perform more detailed/mechanistic studies at the target level, directly in the neuronal cells. The truth is that developments in the field of translational toxicology make it possible to seek complementary information, namely permeability of compounds at the BBB level to, for example, in silico assays [133,134]. Thus, eliminating the need to use endothelial cells as a barrier model, the focus becomes the neuronal response, which also benefits from co-culture with other cells. This point is then explored, providing examples, whenever relevant, of co-culture models with endothelial cells used for neurotoxicity and neurodegeneration research.

The field of translational toxicology brings together epidemiological studies, in silico, in vivo, and in vitro assessments, to quantify and predict the impact of such exposure on the CNS [133,134]. The epidemiological studies allow us to relate, through statistical analysis, the exposure to a chemical to a given disease, thus corroborating the risk of developing pathology and neuronal changes with exposure to potentially toxic compounds [134]. In the translational in silico approach, advances in computational modeling enables to estimate the toxic potential of compounds to cause neuronal damage, based on its structure, physicochemical, and biological characteristics [134]. Another type of translational technique involves in vitro models, through which it is possible to carry out preliminary tests to identify and characterize neurotoxic substances and to perform studies, at the molecular level, to assess mechanisms, biomarkers, and possible pathways for adverse effects [134]. Finally, using animal models, it is possible to perform in vivo studies about neurobehavioral and other qualitative tests to evaluate changes and damage in the CNS, such as biomarker analysis and gene expression [134].

The epidemiological studies fail in providing mechanistic studies to understand how the toxicant affects the cells. On the other hand, mouse, fish, or monkey animal models are used in in vivo neurotoxicity assays, but these models still have limitations related to the interspecies variability (namely in the expression of ion channels and neurotransmitter synthetase). This makes animal models a real challenge to extrapolate the gathered information into humans [133,135]. In the past few years, in vitro techniques to perform risk assessments have emerged. In vitro assays are also popular in drug screening, in studies of drug transport and metabolism and to test drug–drug interactions [26]. In addition, other aspects have been driving the replacement of animal models by in vitro models, namely the fact that in vitro tests allow potentially toxic compounds to be reliably analyzed in more realistic concentrations; decrease in costs, by eliminating the need of animals use; increase in the reliability of safety testing and the legislative pressure to develop alternatives to in vivo testing [135].

When conducting studies to evaluate the neurotoxicity of compounds, it is worth taking into account the complexity of the nervous system itself. Several points must be considered, since they act as facilitators for the induction of neurotoxicity, related to: (i) the lipid-rich structures; (ii) the BBB itself; (iii) the energy requirement; (iv) the synaptic transmission; (v) neural cell structure and, finally, (vi) related to the biochemistry of neurons [135].

As it has been discussed, the BBB limits the crossing of certain substances from blood into the CNS. Nonetheless, the membranes are lipid-based structures and, therefore, they can accumulate lipophilic compounds, such as methylmercury [135]. The neurotoxicity can also be due to certain environmental precursor substances, as 1-methyl-4-phenyl-1,2,3,6-tetrahydropyridine (MPTP), that crosses the BBB and then is metabolically activated by MAO-B present in astrocytes to 1-methyl-4-phenyl-4-phenylpyridinium (MPP+), causing neuronal degeneration [135,136]. This provides an example of how cells that make up the BBB can play an active role in xenobiotic-induced toxicity.

Neurons require ATP in a larger amount to comply with generating and maintaining the membrane potential. Having as a major source of energy the oxidative phosphorylation, neurons become highly vulnerable to inhibitors of the mitochondrial respiratory chain. Examples of toxicants include rotenone and carbon monoxide [135].

The synaptic transmission is a key function of the nervous system and, therefore, a potential target of neurotoxicants. The toxic compounds can target the neurotransmitter release, inhibiting it. A good example is the botulinum neurotoxins. The receptors may also be affected, as it happens for curare and nicotine, both affecting the nicotinic receptors of acetylcholine. At last, the clearance of the neurotransmitters may also be affected (organophosphates inhibit the acetylcholinesterase activity) [135].

Neurons develop very long projections called axons, which are also a recognizable target of toxicants. The substance taxol is known to cause cytoskeleton disruption and impair intracellular transport, leading to peripheral neuropathy [135].

Regarding neuronal biochemistry, it is important to note that dopamine neurotransmitters, produced by dopaminergic neurons, quickly autoxidize. Thus, compounds that increase the release of dopamine, such as methamphetamine, or compounds that act as reuptake/degradation inhibitors (e.g., cocaine), can contribute to increased levels of radicals that damage cells, by intensifying the autoxidation rate of dopamine [135].

Thus, since xenobiotics can have several targets where they can exert their toxic effects, the endpoints to generate more accurate neurotoxicity outputs that must be taken in consideration when performing the assays are related to (i) cell viability/death (e.g., direct counting death cells, apoptosis markers); (ii) evaluation of cell differentiation; (iii) morphological and functional analysis (e.g., cell migration, neurite integrity, glucose uptake, energy metabolism, formation of reactive oxygen species, calcium influx, cell membrane potential); (iv) neuronal specific functions (e.g., electrical activity, neurotransmission); (v) genetic expression profiles, and (vi) neurochemical targets (e.g., enzymatic activity, ion channel function) [134,135,137]. The analytic methods most commonly used for both neurotoxicity and developmental neurotoxicity screenings are spectrophotometric, fluorimetric, and luminometric methods, impedance measurements, electrophysiological screens (e.g., patch-clamp technique), high-content imaging, multichannel parallel microscopy cytometry, transcriptomics, microRNA profiling, and metabolomics [135].

Despite the advantage in bringing the in vitro models progressively closer to the BBB phenotype (including recent advances in microfluidic systems), the use of co-culture models has a more pertinent advantage in this type of assay. By co-culturing cells from the neurovascular unit, it is possible not only to increase the tightness in the model, as well as to reduce the permeability, but also to upregulate the expression of ABC transporters, especially P-gp. As stated before, this upregulation is achieved mostly by culturing astrocytes with endothelial cells (Table 27) [26]. P-gp is a protein from the ABC transporters superfamily, responsible for the efflux of both endogenous and exogenous substances from inside the cells, thereby controlling their accumulation. ABC transporters, including P-gp, present at the BBB, play an important neuroprotective role, by limiting the accumulation of neurotoxicants. Its activity can be induced, activated, or inhibited according to the type of experiment to be performed [23,26]. For example, if the main goal is to evaluate the implication of ABC transporters in drug efflux, when performing a drug permeability study, inhibitors can be used, and the results compared to the non-inhibited conditions. Since efflux transporters show some synergetic effects, studies can be conducted using compounds with affinity for more than one transporter [21,26]. Tariquidar and elacridar are two examples of ABC transporters inhibitors, since both can inhibit not only P-gp but also the BCRP transporter [26]. Using a co-culture-based cell model Megard and colleagues, in 2002, conducted a study regarding the BBB penetration of indinavir [116]. They developed a human in vitro model based on the co-culture of HBEMCs with human primary astrocytes [116]. The results highlighted that the permeability to sucrose of the co-culture model was lower, when compared to the monoculture model of brain endothelial cells. In addition, the co-culture model showed expression of functional P-gp. This result supports the idea that the close contact of astrocytes with endothelial cells gives rise to an improved BBB model to conduct drug permeability studies [116]. Indinavir flux in the apical-to-basolateral direction was low in the co-culture model. When the P-gp inhibitor, quinidine (5 μM for 30 min) was added, there was a significant increase in the amount of indinavir reaching the basolateral compartment in the Transwell system. These results show indinavir as a P-gp substrate, thus confirming P-gp role in preventing its entry into the CNS [116]. In 2013, Ji and colleagues study the influence of dolichyl-phosphate (a glycosyl carrier lipid, playing an important role in the generation of glycoproteins) on P-gp activity, using a co-culture model of RBMECs and primary rat astrocytes [138]. Incubation of the co-culture model with the compound (concentrations of 1–4 µM, for 1 and 4 h) did not significantly change tight junctions, but instead caused a reduction in the transport of rhodamine 123 and Aβ42, both P-gp substrates, from basolateral to apical compartment. This indicated that dolichyl-phosphate inhibits P-gp activity [138].

Co-culture models resorting to microglia cells are also relevant tools in neurotoxicity studies, mainly used to understand the microglia-induced damage to the CNS, once inflammation caused by microglia cells is implicated in structural and functional neuronal changes in this system. For example, lipopolysaccharides (LPS), components from Gram-negative bacteria membrane, and amyloid-β proteins lead the microglia cells to release an inflammatory response to eliminate the trigger responsible for its activation and repair the damage caused to the CNS. However, overactivated microglia cells can cause neuronal damage, including demyelination and even neuronal degeneration [139]. Based on this, Brás and colleagues, in 2020, using a microfluid system (Axon Investigation System) created a co-culture model of microglia (N9 microglial cell line) with hippocampal neurons (from E17 C57BL/6 mice) to evaluate the impact of microglia overactivation on both neuroinflammation and neurotoxicity. TNF-α is a pro-inflammatory cytokine, produced mainly by activated microglia, capable of modulating the most diverse biological responses, from apoptosis to cell differentiation or inflammation [139]. The authors showed an upregulation of miR-342 microRNA in TNF-α-stimulated primary rat microglia. In addition, in the co-culture model, either microglia stimulation with TNF-α (20 ng/mL for 6 h) or microglia transfection with miR-342 dramatically decreased neuronal viability [139]. The following hypotheses were raised by the authors to explain the decreased neuronal viability: (i) increased levels of pro-inflammatory cytokines IL-1β and TNF-α (pointed out as disruptors of the neuronal excitatory/inhibitory ratio); (ii) increased levels of nitrites (microglial-derived nitrites are involved in neuronal respiration inhibition, culminating in excitotoxicity by glutamate release), found in the supernatants of those co-cultures. In conclusion, resorting to the co-culture model, the authors were able to correlate the TNF-α-induced miR-342 with neurotoxicity, through microglia activation [139].

In another example evaluating the neurotoxicity of compounds using monocultures and co-culture models, in 2012, Cardoso and colleagues performed an assay involving monolayers of primary rat microvascular endothelial cells and a co-culture model of primary rat microvascular endothelial cells with astroglial cells [140]. They observed that, following exposure to LPS (1 μg/mL, for 4 or 24 h) or unconjugated bilirubin (50 μM, for 4 or 24 h), both models showed a decrease in TEER values, while their permeability increased. However, the effect was less pronounced in the co-culture model [140]. This was in agreement with studies previously performed, where it was highlighted that the presence of astrocytes helped to fortify the barrier function and helped to prevent endothelial cell permeability, thus concluding that the co-culture model is an improved tool for this type of studies [140]. Another example that includes astrocytes in co-culture is the study conducted by Haruki and colleagues, in 2013. The aim of that study was to identify the mechanism of astrocyte damage in neuromyelitis optica, an inflammatory condition that affects the optic nerves and spinal cord [74]. Based on the discovery of autoantibodies against the aquaporin-4 water channel of astrocytes, the authors used human astrocytes and immortalized them with the temperature-sensitive Simian virus 40 large T-antigen and aquaporin-4 cDNA. The authors designated this astrocytic cell line as hAST-AQP4 [74]. The assays performed, using neuromyelitis optica patients’ sera, consisted of (i) evaluation of the sera effects on both the quantity and localization of aquaporin-4 (cytotoxicity and changes in astrocytes morphology were performed) and (ii) by co-culturing the astrocytic cell line, previously created, with the immortalized cell line TY09, they analyzed the influence of the direct contact with the endothelial cells on the expression and localization of the protein in the astrocytes. The results showed that sera from patients with the pathology effectively induced cytotoxicity in astrocytes, altering their morphology as well as the localization and expression of aquaporin-4 (decreased levels in hAST-AQP4). Regarding the effects of co-culture, the authors verified the positive influence of the direct contact between astrocytes and endothelial cells. Indeed, the expression of the aquaporin-4 in the hAST-AQP4 cells co-cultured with the TY09 cell line was higher, when compared to the monoculture model of the hAST-AQP4 cells [74]. The main conclusion is that the expression and localization of this water channel is influenced by the close contact of astrocytes with endothelial cells, thus demonstrating that this in vitro model has characteristics more similar to those observed in vivo [74].

The SH-SY5Y cell line is another example of relevance in neurotoxicity assessments, in addition to its importance in studies related to neurodegenerative diseases, notably PD [141]. De Simone and colleagues, in 2017, created a co-culture model, aiming to add a relevant in vitro model to the portfolio of in vitro models used for toxicological studies [142]. Using a Transwell system, the model consisted of human neuronal cells (SH-SY5Y cell line) co-cultured with human astrocytes (D384 cell line), and the main goal was to evaluate its neuroprotective role, resulting from cell–cell interaction in co-culture, against three well-established neurotoxicants: methylmercury (1–2.5 μM, for 24 and 48 h), iron nanoparticles (1–100 μg/mL, for 24 and 48 h), and methylglyoxal (0.5–1 mM, for 24 and 48 h). The assay consisted of the analysis of mitochondrial function and cell morphology in the monoculture exposed to the toxicants, comparing the results with exposure to the same compounds in the co-culture model [142]. The three compounds tested were able to cause a significant impairment in the mitochondrial activity and cell morphology in both the monoculture and co-culture models. The release of high levels of iron from the iron nanoparticles is capable of compromising cell viability, as well as iron transportation and storage. Since it is recognized that astrocytes can accumulate iron from exogenous sources, in a time/concentration dependency, the study confirmed that neurons are less susceptible than astrocytes to suffer damage from iron accumulation (results show less iron accumulation in the SH-SY5Y cell line, when compared to the D384 astrocytic cell line). Regarding concerns with methylmercury exposure, the authors confirmed that the neurons showed less alterations when compared to the monoculture model regarding mitochondrial activity and cell membrane integrity. This was also found in another previous study using neurons and astrocytes in co-culture, which was performed by Yin and colleagues, in 2009, and by Morken and colleagues, in 2005 [142]. The main conclusion of this study was that, after exposure, the culture demonstrated enlightened neuronal survival over time. Indeed, the model of neurons co-cultured with astrocytes can diminish the cytotoxic effects of the neurotoxicants in the cells, thus indicating the protective effect of astrocytes to neurons and vice versa, during short exposure to toxicants (24–48 h) [142]. Another example lies in the study carried out by Ferraro and colleagues, in 2020, that, using the SH-SY5Y neuroblastoma cell line, studied the neurotoxicity of titanium dioxide nanoparticles [143]. The cells were exposed to the nanoparticles (5, 10, 50, or 100 μg/mL) for 24 h [143]. The results have shown alterations in: (i) cell viability, as well as the ability of the nanoparticles to induce apoptotic cell death; (ii) reactive oxygen species generation; (iii) antioxidant response (significant nuclear factor erythroid 2- related factor (Nrf2) activation; (iv) endoplasmic reticulum stress, and (v) autophagy in cells. All these results indicated that nanoparticles caused significant neurotoxicity and cell death. Furthermore, based on the main findings, the authors also conclude that more studies are needed to further understand the biological response to titanium dioxide nanoparticles, resorting to SH-SY5Y cells co-cultured with microglia cells, for its relevance in oxidative stress [143].

In 2019, Zhao and colleagues developed an in vitro co-culture model to assess the effect of glutamate-induced excitotoxicity on DNA methylation in astrocytes [144]. Astrocytes express glutamate receptors, which are activated by the glutamate released from excited neurons. Indeed, astrocytes are capable of responding to neuronal activities, since they are the main cells responsible for removing extracellular glutamate, playing a pivotal role in mitigating its excitotoxicity. DNA methylation seems to be responsible for inhibiting the glutamate transporter (EAAT-2) expression in astrocytes [144]. Thus, the researchers created an in vitro co-culture model of primary neurons, astrocytes and endothelial cells from rats, using a Transwell system (neurons were seeded on the bottom of the lower chamber; endothelial cells were seeded on the top membrane, while astrocytes were seeded on the bottom membrane of the insert) and, after the co-culturing different concentrations of glutamate were added and the excitotoxicity and as the DNA methylation was evaluated in astrocytes [144]. The results elucidated that, after glutamate exposure (50, 100, 200, 500, and 1000 μM), the neuronal damage was lower in the co-culture model, when compared to the control group (neurons and astrocytes in monoculture). In contrast, the number of astrocytes was reduced. Concerning DNA methylation, in the co-culture model, both DNA methylation and DNA methyltransferase 1 and 3 expression in astrocytes was increased. In parallel, they examined the effect of MK-801 (10 μM) and 5-AzaC (0.5 μM) in reducing the glutamate-induced excitotoxicity in neurons. The results revealed that the addition of the inhibitors, on the membrane of the Transwell system containing the endothelial cells (to mimic the passage of drugs through to the BBB, thus entering the CNS), was able to efficiently block not only the increased methylation of astrocytes, resulting from glutamate exposure, but also the neuronal damage. Therefore, this model demonstrated to be an improved system to study glutamate-induced toxicity, approximating its characteristics as closely as possible to those found in vivo. The results also allowed to conclude that targeting astrocytes may constitute a new therapeutical strategy in neurological disorders related to an excessive rate of glutamate [144].

One study that highlights the relevance of 3D co-culture models in this type of assessments was published in 2017, where Terrasso and colleagues created a 3D human cell model of neurons and astrocytes to assess the neuroprotective role of 36 compounds (from chemical-synthetized compounds to natural extracts) against oxidative stress caused by chloramphenicol (exposure to 4.3 mM, for 48 h) and tert-butyl hydroperoxide (exposure to 280 μM, for 48 h) [145]. The 3D co-culture model was established by the differentiation of the pluripotent embryonic carcinoma-derived NTera-2/clone D1 (NT2) cell line, using an agitation-based culture system (previously displayed in Section 3.2). After generation of the 3D neuron-astrocyte aggregates, they were collected from the spinner vessels, evaluated for diameter stability, which was kept stable, and then distributed in 96-well plates for cell viability endpoints [145]. They used as the positive control the antioxidant idebenone, which proved to have a dose-dependent neuroprotective effect on the injury induced by tert-butyl hydroperoxide. The result was consistent with previous studies using in vitro cultures of neural cells (primary cortical neurons and immortalized neural cells), in which idebenone also conferred neuroprotection against oxidative stress [145]. The way the model was developed allowed taking into account both cell–cell metabolic and cell-extracellular matrix interactions, mutually biological relevant events to more accurate outputs in toxicity screenings [145]. The results demonstrated the simplicity and applicability of the high-throughput neuroprotection assay, which, combined with the easy evaluation of the fluorescence-based cell viability endpoint, proved to be a suitable model for drug screening assays [145].

### 3.5. The Use of In Vitro Co-Culture Models in the Study of Neurodegenerative Diseases

In addition to their relevance in neurotoxicity studies, in vitro cell-based models have also been widely used in neuroscience studies related to neurodegenerative diseases [146]. Indeed, in vitro BBB modeling can be used for three major categories of studies: toxicological evaluations, BBB transport studies, and target validation at the disease mechanism as well as its therapeutical implications [147]. The potential of many promising compounds to act on the CNS is diminished by the presence of the BBB [147]. Moreover, the BBB is implicated in neurodegenerative diseases. Its dysfunction in various pathologies may impair the transport and permeability of substances, causing alterations in the regulatory mechanisms between endothelium and its associated cells, such as neurons and glia cells [147]. Thus, by modeling the BBB it is possible to make not only an uptake prediction of potential new candidates for the treatment of CNS pathologies, but also to assess the risk of using such compounds in pharmaceutical development, generating information with impact on drug design [147].

Historically, only neurons have been considered in in vitro studies designed to develop new therapeutic strategies for neurodegenerative diseases. Recent studies have elucidated the importance of astrocytes in maintaining a proper neuronal function. In reality, astrocytic alterations have been implicated in the pathogenesis of neurodegenerative diseases, such as Alzheimer’s, Parkinson’s, and Huntington’s diseases, and amyotrophic lateral sclerosis (ALS) [146,148]. Perturbations in protein processing, trafficking, and aggregation have been implicated in the pathogenesis of these diseases [149]. This inability to maintain protein homeostasis results in extracellular accumulation and, consequently, in the induction of local inflammation due to the activation of astrocytes and microglia [149]. Thus, considering the key biological functions of astrocytes, their dysfunction may contribute to the pathogenesis of neurodegenerative diseases [8,148]. For example, in Alzheimer’s disease (AD), large accumulation of tau protein in astrocytes has been associated with the diminished expression of glutamate transporters [148]. Considering that dysfunctional glutamate neurotransmission has been associated with neuronal death, this supports a role for astrocytes in the pathogenesis of neurodegenerative diseases. [149]. The role of microglia cells in neurological disorders is related to their double phenotype, pro-inflammatory phenotype M1 and anti-inflammatory phenotype M2 [146]. Indeed, some inflammation-inducing factors are associated with neurodegenerative diseases, being responsible for the amplification of the inflammatory response with the production of neurotoxic mediators, such as cytokines and interleukins. Those neurotoxic mediators are commonly related to intracellular mechanisms described in these pathologies, such as protein degradation, mitochondrial and axonal dysfunction, and apoptosis [150].

Enright and colleagues, in 2020, performed an assay that was able to elucidate the advantage of co-culture models in obtaining neuronal responses as close as possible to the in vivo conditions [151]. Resorting to a microfabricated multi-electrode array (MEA), they created a complex model of primary rat cortical neurons co-cultured with primary rat astrocytes and oligodendrocytes to perform morphological, functional, and transcriptional analysis. Like astrocytes, oligodendrocytes secrete certain factors responsible for intervening in the maturation of neurons, as well as in processes like synaptogenesis. In addition, they are also able to provide metabolic support and to promote myelination of axons, thus enhancing both the speed and efficacy of nerve conduction [151]. Making the comparison between a neuronal monoculture and the co-culture system (neurons, astrocytes, and oligodendrocytes), they showed that the co-culture system presented (i) a high number of regions with higher neuronal density and differences in synaptophysin expression; (ii) a higher expression of genes involved in synapses, network formation, and cell signaling processes, and (iii) a higher neuronal activity, as well as better synchronization on the electrophysiological activity measurements. The authors concluded that the co-culture model can mimic more closely the in vivo neuronal responses by taking into consideration the cell–cell interactions, therefore providing an improved tool for testing drugs, toxicants, or even disease mechanisms [151].

In 2021, Mursaleen and colleagues developed an in vitro co-culture model of immortalized endothelial cells, hCMEC/D3 cells, with the neuroblastoma cell line, SH-SY5Y that elucidates the BBB modeling for drug delivery and target validation [152]. In in vivo studies, hydroxytyrosol has shown a low capacity to cross the BBB. To overcome this problem, the authors used micellar nanocarriers to enhance the passage across the BBB and used rotenone (concentration of 100 µM) to mimic a PD phenotype [152]. They showed that micellar nanocarriers containing hydroxytyrosol were capable of crossing the BBB, in vitro, without causing cytotoxicity, and protected the neuronal SH-SY5Y cells against rotenone-induced oxidative stress (assessing by mitochondrial hydroxyl levels) [152].

In the last part of this review, we will address studies that made use of co-culture models to explore neurodegenerative disease mechanisms. We will focus on the three most prevalent neurodegenerative diseases, Alzheimer’s and Parkinson’s diseases, and amyotrophic lateral sclerosis. These three neurodegenerative disorders have in common the fact that both incidence and prevalence increase with age and, although some cases are familial, most cases are sporadic with onset at early age [153,154].

#### 3.5.1. Alzheimer’s Disease

Alzheimer’s disease is a progressive neurological disorder, leading to loss of cognitive functions, changes in social behavior and basic tasks impairment, such as memory, swallowing, and walking [155,156]. The progressive neurodegeneration is accompanied mostly by the accumulation of senile plaques (extracellular aggregates of amyloid-β peptides) and neurofibrillary tangles (intracellular inclusions of tau proteins) [157]. Moreover, other pathological signs have been described, as inflammation, BBB disruption, metabolic disruptions, degraded cellular pathways, and increased DNA damage [156,157].

In 2017, Mehrabadi and colleagues used a co-culture model to analyze if microglia cells can affect the BBB integrity. Since microglia cells closely interact with endothelial cells, they can influence the infiltration of peripheral cells by releasing cytokine/chemokine molecules [158]. In a previous study made by the same group, they identified the presence of PARP-1 as necessary to regulate the microglia activation, the release of NOS/ROS and the production of pro-inflammatory cytokines [158]. The immortalized mouse brain endothelial b.END3 cell line was co-cultured with primary microglia cells from rats. The microglia cells were exposed to amyloid-β and/or PARP-1 inhibitor PJ34 (5 µM for 24 h) for trigger microglia activation and the expression of ZO-1 and occludin in endothelial cells was then evaluated. Amyloid-β-exposed microglia reduced the expression of the tight junction proteins ZO-1 and occludin by co-cultured endothelial cells, an effect fully prevented when co-cultured microglia cells were treated with amyloid-β and PARP-1 inhibitor [158]. The reduced expression of tight junctional proteins and the consequent increase in the paracellular permeability were further studied by the determination of the microglia phenotype. The pro-inflammatory M1 phenotype is described to cause the BBB opening. Thus, the authors investigated the presence of phenotype biomarkers for BBB opening, such as nitric oxide and TNF-α, both present after amyloid-β exposure. The confirmation of the ability of these compounds to alter the junctional proteins in the endothelial cells was also assessed [158]. Exposure to sodium nitroprusside (source of nitric oxide, concentrations of 20 e 40 µM) reduced occludin expression but did not affect the ZO-1 expression. Meanwhile, exposure to TNF-α (30 and 50 ng/mL) decreased the ZO-1 expression, with no changes in the occludin expression [158]. Exposure to both molecules caused an increase in the permeability of fluorescein-labeled dextran and IRDye 800 PEG. This co-culture model of endothelial cells with microglia elucidated the effects of cell–cell communication, allowing the authors to conclude that the microglia can interfere with endothelial cells barrier function and the deleterious effects of amyloid-β-activated microglia are dependent on PARP-1 activity [158].

Related to the impact of energy metabolic disruption in AD, Hong and colleagues, in 2020, investigated the potential role of the SLC transporter monocarboxylate transporter-4 (MCT-4) as a putative target in the treatment of AD [155]. The authors performed in vitro and in vivo experiments. The in vitro experiments were performed in a co-culture model of primary neuronal cells and astrocytes from the hippocampi of day 16 embryos of APP/PS1 mice [155]. In astrocytes, the intracellular transportation of lactic acid is made via MCT-4. It was shown that: (i) increased expression of MCT-4 in astrocytes stemmed in the accumulation of lactic acid in the astrocyte’s stroma, resulting in acidosis; (ii) as a consequence of MCT-4 overexpression, neuronal proliferation was diminished and the apoptosis rate was enlarged; (iii) higher content of lactic acid in stroma increased neuronal oxidative phosphorylation; (iv) acidosis can lead to mitochondrial dysfunction [155]. Therefore, the changes in the co-culture model, involving the MCT-4 transporter, compromised the neuronal mitochondrial function, lactate metabolism and apoptosis rate. In the in vivo experiment, after knocking down the expression of MCT-4 in APP/PS1 mice, they verified that the apoptotic rate of neurons was reduced, and the cognitive function of mice was improved [155]. By combining the results obtained in vitro, in the co-culture model, with the results of the intervention on transporter expression in vivo, it was possible to establish a correlation between the energy metabolism of astrocytes and AD pathogenesis [155].

In 2018, Gupta and colleagues also performed an assay related to the link between metabolic impairments and sporadic AD, more specifically insulin signaling disfunction, a risk factor leading to disturbances in learning and memory [159]. In that study, they analyzed the metabolic status of glial and neuronal cells and tried to understand how the crosstalk between these cells can influence the progression of the disease [159]. The intracerebroventricular administration of streptozotocin (a non-transgenic metabolic model) causes abnormalities in insulin signaling/glucose metabolism. In fact, streptozotocin is a compound able to demarcate the role of insulin signaling in the brain, allowing us to study its link with amyloidogenesis and tau pathology [159]. They resorted to a streptozotocin-induced glial-neuronal co-culture model (cells were exposed to 100 μM of the compound for 24 h), in plates, by co-culturing the rat astrocytoma cell line C6 with a neuronal rat pheochromocytoma cell line, PC12. In parallel, they also developed an in vivo streptozotocin-induced rodent model to extrapolate the results from the in vitro model [159]. In the monoculture model, glial cells show a significant increase in amyloid precursor protein (APP) and β-secretase1 (BACE1) protease, while neurons did not. Meanwhile, in the co-culture model, neurons showed an increase in APP and BACE1 transcripts that actually exceed those found in glia. This result highlights the neuronal vulnerability in the presence of glia. Also in the co-culture model, amyloid-β and BACE1 expression were significantly increased. Related to protein tau, similar results were discovered in the in vivo and in vitro model. Another finding that highlights neuronal vulnerability in the presence of glial cells was the significantly reduced expression of glia-specific GLUT-1 and neuronal GLUT-3 transporters in the co-culture model, but not in the monoculture system. Decreased GLUT-1 expression was related to impairments in glucose uptake in the co-culture, with major changes observed in glia, since insulin signaling is coupled to the glucose uptake in astrocytes. In the animal model, changes in GLUT-1 and 3 were also found, with a significant decrease in the transcripts of both transporters [159]. This suggest that streptozotocin induces insulin signaling dysfunction, a mediator correlated with sporadic AD pathology. More importantly, the co-culture model allowed the drawing of the inference of astrocytes in mediating pathological responses in neuronal cells [159].

In 2017, Spampinato and colleagues created an in vitro BBB co-culture model of human endothelial cells (using the TY10 cell line) and human astrocytes to assess the effect of cell–cell communication in the presence of amyloid-β accumulation in endothelial barrier. Cell cultures were exposed to amyloid-β peptide, namely Aβ1-42 (2 μM for 18 h) [78]. They showed that astrocytes can influence amyloid-β effects towards endothelial cells. While TY10 cells in monoculture exposed to amyloid-β peptide showed no differences in dextran permeability, dextran diffusion, accompanied by modifications in the structure of the tight junction claudin-5, were revealed in the co-culture model of TY10 cells with astrocytes. Since amyloid-β exposure was not able to provoke cell death in either endothelial cells or astrocytes, the hypothesis that the sensitivity demonstrated by the endothelial cells could be due to soluble factors released by astrocytes was raised. The authors verified that exposure to the peptide induced the expression of VEGF (VEGF mRNA and protein levels were enhanced). VEGF plays a role in angiogenesis and, furthermore, is released by active astrocytes when stimulated by inflammatory processes. As a result, VEGF induces a down-regulation of tight junctions [78]. In addition, MMP-9 was also found to be increased in endothelial cells in the co-culture model. This finding is supported by others in vitro studies that implicated MMPs in the degradation of BBB tight junctions, whose activity is induced by the presence of amyloid-β peptide. The modification in the barrier function of endothelial cells was prevented upon addition of the non-specific MMPs inhibitor, GM6001, to the co-culture, thus validating the finding [78]. In order to establish the link between astrocyte-derived VEGF and MMP-9 with its implications for altering endothelial permeability, the same experiments were performed, with the amendment of VEGF down-regulation in astrocytes. The findings highlighted an attenuation in the effects caused by VEGF in both the expression of MMP-9 and in the maintenance of the barrier effect. This led the authors to conclude that astrocyte-mediated activity may play an important role in maintaining the barrier provided by endothelial cells [78]. In 2019, the same group, resorting to a co-culture model of a human immortalized endothelial cell line (TY10 cells) with human astrocytes, studied the influence of astrocytes regarding leukocyte migration through the BBB in the presence of amyloid-β protein [160]. Indeed, the disruption of the BBB in AD is known to increase the permeability of immune cells, responsible for participating in the inflammatory reply. Therefore, the TY10 endothelial cells (monoculture) and the TY10 endothelial cells co-cultured with the human astrocytes were exposed to amyloid-β (2.5 µM), to inflammatory conditions (TNF-α and IFN-γ, 10 U/mL and 5 U/mL, respectively) or to the combination of both, for 5 and 18 h [160]. They showed a significant endothelial cell death in the monoculture model after 5 h under inflammatory conditions (amyloid-β induced cell death after 18 h treatment). In contrast, in the co-culture model, the inflammatory conditions did not induce endothelial cell death after 5 h of treatment, but instead after only 18 h of exposure [160]. Faced with these results, the co-culture model was used for further studies, with exposure conditions to amyloid-β, inflammatory conditions (TNF-α and IFN-γ) or the combination of both, for 5 h [160]. Concerning permeability evaluations, the dextran permeability was significantly enhanced, and both the claudin-5 and VE-cadherin expressions were reduced, in all three exposure conditions. In the monoculture, amyloid-β exposure induced leukocyte migration across the BBB, while no significant changes in leukocyte migration were induced by the inflammatory cytokines [160]. Additionally, in the co-culture model, the leukocyte migration was only detected under inflammatory conditions. In fact, exposure of mono- and co-culture models to inflammatory conditions for 5 h increased the expression of the adhesion molecule, ICAM-1, which is involved in facilitating the transmigration of peripheral blood mononuclear cells through the BBB. Thus, astrocytes play a role in influencing endothelial cell barrier function, which may happen throughout the release of soluble mediators [160].

Dos Santos Rodrigues and colleagues, in 2019, created an in vitro co-culture model of the BBB (immortalized mouse brain endothelial cell line b.End3 co-cultured with primary rat glial and neuronal cells) in order to assess, via transferrin receptor, the targeting and transfection effectiveness of liposomes, designed to evaluate a potential gene therapy for AD that targets the apolipoprotein E2 [161]. The function of the apolipoprotein E, in the CNS, consists of transporting cholesterol from the astrocytes to the neurons. By controlling cholesterol homeostasis, apolipoprotein E contributes to synaptic plasticity and neuronal function. Three polymorphic alleles are described (ε2, 3, and 4), although only the ε2 allele presents neuroprotective functions (implicated in amyloid-β neuronal clearance) [161]. Higher values of TEER were achieved in the co-culture model of endothelial cell line b.END3 with glial cells, when compared to the monoculture [161]. After TEER evaluation, neuronal cells were seeded at the bottom of the well plate. The ability of liposomal formulations to cross the BBB and transfect the primary rat neuronal cells was demonstrated (100 nM of the liposomal formulations and incubation for 8 h) [161]. In vivo, in C57BL/6 mice, the accumulation of liposomes within the mice brains increased apolipoprotein E2 protein expression [161]. In conclusion, the co-culture model has proven to be an important improved tool in this study, since it was able to assist in both brain target characterization and translocation of liposomal gene carriers across the BBB [161]. In another study involving apolipoprotein E in AD, Topal and colleagues, in 2020, assessed the endothelial permeability of lipid nanoparticles containing donepezil in a co-culture model of BBB [162]. To successfully increase the amount of nanoparticles reaching into the brain, targeting ligands on their surface are usually used. Some peptides and proteins necessary for maintaining normal brain function, namely insulin, transferrin, and lipoproteins, cross the BBB via receptor-mediated transcytosis [162]. In association with lipids, apolipoprotein E forms lipoproteins, having high affinity to innumerous low-density lipoprotein-associated receptors (LRPs) and low-density lipoprotein receptors (LDLRs). Since these receptors are abundant in endothelial cells and neurons, apolipoprotein E becomes a good candidate to target ligands to increase drug delivery into the CNS [162]. To investigate the efficiency of apolipoprotein E-targeting for delivery donepezil across the BBB, the authors developed a co-culture model of primary rat endothelial cells with primary rat astrocytes and pericytes. In parallel, to perform cellular uptake studies, the authors resort to primary rat endothelial cells, hCMEC/D3 cells, and SH-SY5Y cells, differentiated into a more dopaminergic phenotype [162]. The results showed that the uptake of nanoparticles with the apolipoprotein E targeting ligand was higher than those without it, with higher uptake values detected in primary endothelial cells, followed by hCMEC/D3 cells and differentiated SH-SY5Y cells [162]. The permeability assay was performed in the co-culture model and, in addition to TEER measurements, the integrity of the model was also assessed with the evaluation of the permeability values of albumin and sodium fluorescein. The permeability results for both molecules were very low, thus reflecting a good barrier tightness [162]. These results also highlighted a good permeability across the BBB for the targeting nanoparticles. All these results led the authors to conclude that, although further studies in a human BBB model are needed, the targeting nanoparticles can indeed help to increase not only the donepezil crossing into the brain, but also to increase its uptake from neurons, thus increasing drug therapeutic efficiency [162].

As an example of the application of co-culture models in the evaluation of the therapeutic potentiality of compounds, in 2018, Alvariño and colleagues assessed the capability of two new amphiphilic glycosylated angucyclinones isolated from Streptomyces sp, streptocyclinones A and B, to attenuate some impairments related to AD. For this purpose, they used a co-culture model of neuroblastoma SH-SY5Y cells and BV2 murine microglial cells [163]. Regarding the neuronal cell line, after oxidative stress induction with hydrogen peroxide (150 μM for 6 h), the two streptocyclinones reduced the formation of reactive oxygen species. The authors verified that both molecules induced an increase in glutathione levels and the Nrf2 expression in the nucleus, when compared to the control cells [163]. In the microglia cell line, the ability of the compounds to modulate neuroinflammation was also tested, after exposure of BV2 cells to LPS (500 ng/mL for 23 h). Streptocyclinones reduced the release of pro-inflammatory factors and, additionally, induce the translocation of Nrf2 to the nucleus of microglial cells [163]. Following the results that highlighted the neuroprotective role of streptocyclinones, a Transwell co-culture model was assembled (microglial cells were seeded in the inserts above the neuroblastoma cells) to test the protective effect of streptocyclinones on neuronal cell survival rate following LPS-induced microglial cell activation. The viability of neuronal cells increased upon the exposure of BV2 cells to the compounds. Thus, the in vitro assessments allowed the authors to establish a neuroprotective and antioxidant capability for both compounds, by modulating the inflammatory behavior of microglia cells [163].

Knowing that the BBB is a remarkable obstacle for the intracranial delivery of therapeutic drugs for neurodegenerative diseases, such as AD, Wasielewska and colleagues developed, in 2022, a sporadic AD BBB model for the ultrasound-mediated delivery of aducanumab and anti-tau antibodies [164]. In this context, focused ultrasound combined with microbubbles (FUS^+MB^) was used, a novel approach to temporarily disrupt the BBB and enhance drug delivery. From apolipoprotein E gene allele E4 (APOE4, high sporadic AD risk) and allele E3 (APOE3, lower AD risk) carrying patient-derived iPSCs, BBB cells [induced brain endothelial-like cells (iBECs)] and astrocytes (iAstrocytes) were generated to establish monocultures and co-cultures of human sporadic AD and control BBB cells, aiming at assessing the effects of FUS^+MB^ on BBB cell phenotype. These models were also used to screen the delivery of two potentially therapeutic AD antibodies: an Aducanumab analogue (Aduhelm^TM^; anti-Aβ antibody) and a novel anti-tau antibody, RNF5 (an anti-tau antibody that effectively reduces *p*-tau levels in an animal model with tau pathology). Interestingly, a significantly increased delivery (up to 1.73-fold) of RNF5 and Aducanumab analogue was observed across the Transwell-based BBB models upon FUS^+MB^ treatment, when compared to untreated cells. Outstandingly, the obtained results also clearly demonstrated the safety of the FUS^+MB^ approach, as evidenced by minimal alterations in iBEC transcriptome and negligible or no changes in iBEC or iAstrocyte viability and inflammatory responses within the initial 24 h post FUS^+MB^ treatment. In this study, a novel hydrogel-based 2.5D BBB model was also developed as a preliminary step towards a more physiologically relevant FUS^+MB^ drug delivery platform. A successful iBEC barrier formation was observed in the novel 2.5D hydrogel-based BBB model, which led to a significant increase (1.4-fold) in the delivery of the Aducanumab analogue following FUS^+MB^ treatment. Overall, these results highlighted the successful development of a robust and reproducible in vitro model using patient cells for an efficient FUS+^MB^-mediated drug delivery screening. This novel cell-based platform for FUS^+MB^ drug delivery has a remarkable potential in the identification of new FUS^+MB^-deliverable drugs and for the screening of the specific effects of a FUS^+MB^ approach in control- and patient-derived cells, which could accelerate the use of FUS^+MB^ as a therapeutic approach in AD [164].

Given that inflammation is a recognized risk factor for neurodegenerative diseases such as AD, apart from also contributing to its progression, a fully human in vitro BBB model was developed aiming to better understand how inflammation impacts the properties of BMECs, cells that constitute the BBB, which is compromised in AD [165]. The developed model featured BMECs derived from induced pluripotent stem cells and astrocytic cells obtained from differentiating neural stem cells (NSCs) and aimed to investigate the effects of neuroinflammation on barrier function. To mimic neuroinflammation, cells were exposed to recombinant human TNF-α (10 ng/mL), IL-6 (10 ng/mL), or 10 ng/mL each, for 24 h, either in monoculture or in co-culture. In iPSC-derived BMEC monocultures, the TNF-α and IL-6 cytokines were directly responsible for a dysfunction of the BBB, resulting in an impaired barrier integrity, as evidenced by a decreased TEER value (measure of barrier integrity), an increased permeability of sodium fluorescein (a measure of paracellular permeability), and a decreased cell polarity. Indeed, although the TEER remained unchanged upon exposure to IL-6 alone, a significant reduction by 13% and 16% was observed after exposure to TNF-α and both cytokines, respectively. Furthermore, sodium fluorescein permeability increased by 2- to 2.5-fold upon exposure to the inflammatory cytokines, while the efflux ratio of the P-gp substrate rhodamine 123 (ratio between the amount transported out of the brain compartment and the amount transported into the brain compartment) decreased in the presence of the inflammatory cytokines (approximately 40% lower than in the absence of cytokines), evidencing the loss of polarity of BMECs. Overall, all these findings allow to establish a connection between neuroinflammation and specific aspects of BBB integrity impairment. Additionally, either BMECs grown in monoculture or co-cultures of BMECs with NSC-derived astrocytic cells were exposed to TNF-α and IL-6 (10 ng/mL each) and, 24 h later, 25 cytokines and chemokines were quantified in the abluminal media and compared to cultures grown in the absence of the inflammatory cytokines. Increased levels of several pro-inflammatory cytokines [namely, monocyte chemotactic protein 1 (MCP-1), interleukin 8 (IL-8), interferon gamma-induced protein 10 (IP-10), macrophage inflammatory protein 1 beta (MIP-1β), interleukin 1 beta (IL-1β), monokine induced by gamma interferon (MIG)(), and regulated upon activation, normal T cell expressed and presumably secreted (RANTES)], which are typically secreted by astrocytes or endothelial cells, reached their peak during inflammatory conditions in the presence of NSC-astrocytic cells. Outstandingly, and despite the presence of several pro-inflammatory cytokines known to impair the BBB function, NSC-derived astrocytic cells were able to counteract the effects of inflammation, as evidenced by the restoration of TEER values and IgG permeability (a second measure of barrier integrity). Furthermore, these findings also indicate that, in neuroinflammation, a breakdown in transcellular transport seems to occur before an impairment in paracellular permeability. The developed co-culture model effectively replicated the cellular responses to inflammation at the BBB and may represent a useful platform for studying the contributions of individual cell types to disease progression. Furthermore, these findings also suggest that BBB proper functioning may be influenced by a delicate balance of soluble factors, underscoring the intricate nature of inflammation in neurodegenerative diseases, such as AD [165].

Microglia, as brain-resident macrophages, play crucial roles in maintaining homeostasis and create a supportive environment for neurons. Furthermore, microglia are increasingly recognized as playing a significant role in brain pathology, particularly in neurodegenerative diseases such as AD, PD, and motor neuron diseases (MND), with numerous genes associated with these conditions being expressed in microglia. As such, there is a critical need for authentic and efficient in vitro models to investigate the pathological mechanisms of human microglia [166]. To accomplish with this need, Haenseler and colleagues developed, in 2017, an iPSC microglia co-culture model with iPSC cortical neurons. In the developed co-cultures, the neuronal maturity and functionality were retained for extended periods. Furthermore, microglia in co-culture expressed essential microglia-specific markers (namely, MERTK, GPR34, PROS1, C1QA, GAS6, and P2RY12), as well as genes relevant for major neurodegenerative diseases [namely, AD-related genes (*APP*, *PICALM*, and *CD33*); PD-related genes (*PARK15*, *PINK1*, *SNCA*, and *DJ*-1); and MND/amyotrophic lateral sclerosis-related genes (*C9orf72*, *TDP43*, and *SOD1*)], suggesting that the developed co-culture model could be useful to investigate the effects of several genes associated with highly debilitating neurodegenerative diseases. Furthermore, iPSC-derived microglia in co-culture also exhibit highly dynamic ramifications and demonstrate phagocytic activity. Overall, given the increasing evidence of the implications of microglia in distinct neurodegenerative diseases, the developed iPSC-derived microglia model, by expressing key genes involved in AD, PD, and MND, could represent a useful tool for the study of the corresponding gene products within an authentic human in vitro system. Noteworthy, some of these disease-relevant genes can be studied in microglia either in monoculture or in co-culture conditions, being the developed co-culture conditions especially important for genes where the crosstalk between microglia and neurons is essential [166].

#### 3.5.2. Amyotrophic Lateral Sclerosis

Amyotrophic lateral sclerosis (ALS) is an uncurable neurodegenerative disorder characterized by a progressive degeneration of both upper and lower motor neurons (brain and spinal cord), culminating in death within a short time after diagnosis. The cause of death is usually respiratory failure [167,168]. The pathology of this disease also involves cell death dysregulation [167].

Smethurst and colleagues, in 2020, based on a pathological finding for this disease, the aberrant cytoplasmic localization and aggregation of TAR DNA binding protein 43 (TDP-43) in motor neurons and glia (in both brain and spinal cord of patients), performed an in vitro assay to further analyze TDP-43 toxicity in cells and to try to understand the role of astrocytes in ALS [168]. They used a co-culture model of human induced pluripotent stem cell-derived motor neurons and astrocytes to modeling cell-specific characteristics of sporadic ALS. For TDP-43 seeding aggregation it was used serially passaged ALS spinal cord extract [168]. In the monoculture model of derived motor neurons, the researchers have noticed, after treatment with seeding aggregation, a lack of toxicity, accompanied with low frequency and abundance of TDP-43. Meanwhile, combined seeded aggregation with proteasomal inhibition significantly increased the number of aggregates and activated caspase-3. This finding suggested that TDP-43 seeded aggregation is toxic only under defined cellular conditions. In the co-culture model, TDP-43 was toxic to neurons but not to astrocytes. In addition, astrocytes demonstrated a neuroprotective role by reducing TDP-43 aggregation and toxicity in neurons. This model supports the clinical relevance of the protein-mediated toxicity hypothesis in ALS, raising the importance of astrocytes in this disease [168].

Another study highlighting the importance of cell–cell communications between astrocytes and neurons was conducted by Paul and Belleroche, in 2014 [169]. D-serine (a co-agonist of N-methyl-D-aspartate receptor) levels are elevated in sporadic ALS and a pathogenic mutation on the enzyme responsible for degrading D-serine, D-amino acid oxidase, co-segregates with familial ALS. The pathogenic mutation R199w in D-amino acid oxidase severally impairs enzyme activity, stimulating toxic effects in motor neurons, resulting in apoptosis. Based on these findings, the authors used a motor neuron cell line, NSC-34 cells, and the C6 glioma cell line in both monoculture and co-culture (Transwell system) to analyze the toxicity mechanism of the mutant protein and the interactions between both cells [169]. Monocultures of C6 cells expressing wild-type or R199w- D-amino acid oxidase mutation showed low levels of apoptosis. In a co-culture model of a C6 cell expressing the mutant version of D-amino acid oxidase, NSC-34 cells experienced an increased staining for the apoptotic marker annexin-v, when compared to the co-culture containing C6 cells expressing the wild type version of D-amino acid oxidase. This suggested that a dysfunction in D-serine metabolism may cause a release of some component(s) from glial cells that regulates neuronal viability. Blocking of the glycine/D-serine binding site at N-methyl-D-aspartate receptor with the selective antagonist 5,7-dichloro-4-hydroxyquinoline-2-carboxylic acid (DCKA) significantly attenuated the neuronal apoptosis caused by the mutant version of D-amino acid oxidase [169]. In conclusion, the authors showed that elevated levels of D-serine play a role in motor neuron degeneration and highlighted the importance of cross-talk between cells in disease pathology [169].

To better understand the disease physiology, Veyrat-Durebex and colleagues, in 2016, developed an in vitro model of ALS, using a Transwell system, resorting to a motor neuron-like cell line, NSC-34 cells, co-cultured with the astrocyte clone C8-D1A. Both cell types were over-expressing the human wild-type or G93C mutant superoxide dismutase-1 (SOD-1) [170]. The analysis focused on the role of SOD-1 mutation and oxidative stress (induced by 10 μM of menadione) on intracellular metabolism, both implicated in ALS pathogenesis [170]. Under oxidative stress, metabolomic analysis showed alterations in glycine, succinate, malate, and glutamate concentrations in the model over-expressing mutant SOD-1. This suggests a disruption in tricarboxylic acid cycle, in excitatory neurotransmission and in glutathione synthesis [170]. The implications of the impairment of these metabolic pathways are related to excitotoxicity and dysfunction in mitochondrial respiratory chain activity, both involved in neuronal injury. These findings are consistent with prior research in ALS. Therefore, the co-culture model shows to be a well-designed and useful model, allowing the assessment of the consequences of oxidative stress on energetic metabolism in ALS [170].

Lee and colleagues, in 2017, used a co-culture model of b.END3 cells with primary mice astrocytes to assess the influence of endothelial cells on the expression of glutamate transporter, GLT-1 by astrocytes [171]. GLT-1 is a sodium-dependent transporter, responsible for glutamate clearance that shows a reduced expression in postmortem samples of ALS patients. They showed that the presence of endothelial cells in the co-culture model increased GLT-1 expression by astrocytes, as compared to astrocytes in monoculture [171]. This effect was attenuated by γ-secretase inhibition with DAPT (10 μM) and by knocking-down the notch effector, RBPJ. This indicates that notch signaling, which occurs as a result of cell contact, is pivotal to generate an up-regulation of astrocytic GLT-1 by endothelial cells [171].

Mohamed and colleagues, in 2019, studied the influence of glutamate release by the astrocytes in promoting P-gp expression in endothelial cells, and its consequence in ALS by reducing the CNS penetration and efficacy of the ALS drug riluzole [172]. The authors developed an in vitro human BBB model by co-culturing patient-derived induced pluripotent stem cells differentiated into endothelial cells with astrocytes differentiated from patient-derived induced pluripotent stem cells of SOD1-A4V or C9orf72 genotypes [172]. They showed that endothelial cells exposed to the conditioned media from SOD1-A4V-derived astrocytes presented an increased expression and activity of P-gp, via NMDA receptor activation [172]. In contrast, exposure to conditioned media from C9orf72 astrocytes did not produce the same result, which may indicate that this phenomenon is not shared by all forms of ALS [172]. COX2 inhibition is suggested as a promising therapeutic target in overcoming glutamate-mediated P-gp upregulation in epilepsy and PD animal models [172]. Based on this, they next blocked the NMDA receptors and COX2 with MK801 and celecoxib, respectively. Both inhibitors significantly diminished P-gp expression in endothelial cells exposed to the conditioned media from SOD1-A4V-derived astrocytes. They concluded that glutamate released by SOD1-A4V and sporadic ALS astrocytes mediated P-gp upregulation in endothelial cells of the BBB and revealed possible new targets for ALS treatment [172].

#### 3.5.3. Parkinson’s Disease

Parkinson’s disease is a neurological disorder mostly characterized by the degeneration of dopaminergic neurons [173].

As previously mentioned, SH-SY5Y cells are capable of being differentiated into a dopaminergic phenotype [88,89]. Therefore, this neuroblastoma cell line is a widely used model for PD research. There are two main strategies used to induce the disease phenotype, including drug treatment and/or genetic approaches. The compounds used for this purpose, based on drug-induced phenotype, are the MPP+ (toxic metabolite formed from its precursor MPTP, a product used in the synthesis of a synthetic analog of heroin, responsible for causing severe Parkinsonism in humans); 6-hydroxydopamine (catecholaminergic neurotoxin), and rotenone (highly lipophilic insecticide). The mechanism by which these compounds induce neurotoxicity mainly involves mitochondrial dysfunction and oxidative stress induction [89]. In a recent work, using a 3D scaffold, Chemmarappally and colleagues, in 2020, developed a co-culture system of the SH-SY5Y neuroblastoma cell line (differentiated into a more mature dopaminergic phenotype with retinoic acid and B27 supplement), with the astrocytic cell line (U-87MG human glioblastoma cell line) [141]. Indeed, neurons and astrocyte-based cell models have been used in neurodegenerative diseases and CNS disorders studies, since astrocytes have been involved in neuroprotection by releasing antioxidant molecules and neurotrophic factors (neurons can use the antioxidants released by astrocytes to reduce the oxidative conditions) [141,174]. Neuronal cells showed an improved survival rate when co-cultured with astrocytes in the 3D scaffolds, after challenging the system with rotenone, MPTP, MG132 (induces proteasome inhibition) and BSO (glutamyl-cysteine synthetase inhibitor) [141]. Furthermore, when compared to 2D or even 3D monoculture models, the developed model has the advantage of allowing the neuronal cells to communicate more naturally with neighboring cells, providing a protective effect. Thus, the authors concluded that scaffolds could be an improvement to the models used for drug discovery in PD, as well as in studies related to disease progression [141].

In 2015, Efremova and colleagues created a co-culture model of human dopaminergic neurons (using LUHMES cells, conditionally immortalized mesencephalic neuronal precursors) and astrocytes (immortalized mouse astrocyte cell line, IMA 2.1) to assess the neuroprotective role of several compounds against the neurotoxin MPTP [173]. Astrocytes are responsible to convert MPTP into its toxic metabolite MPP+. In fact, conversion of MPTP requires monoamine oxidase (MAO) activity, mainly found in astrocytes. Therefore, taking advantage of the fact that the astrocytic cell line was used to create the in vitro model, they also analyzed possible differences to the direct use of MPP+ in in vitro models, which omit the influence of glial cells in the process [173]. The co-culture model recapitulated the in vivo conversion of MPTP into MPP+ and its inherent neurotoxicity. In a time-dependent manner, a decrease in MPTP levels in the supernatant was verified, accompanied by the formation of MPP+, reaching saturation levels after 24 h [173]. Co-cultures treated with MPTP (30 μM) for 5 days revealed a noticeable neuronal loss, as detected by β-III-tubulin staining. For pharmacological evaluations, authors compared the MPP+ model (monoculture of the neuronal cell line) and the MPTP model (co-culture). DAT blockage with GBR12909 showed a neuroprotective role in both models, while MAO-B inhibition with L-deprenyl only prevented the toxicity of MPTP in the co-culture model. These results support the data from in vivo studies, demonstrating that MPTP is converted by MAO into MPP+ in astrocytes [173]. In In general, the experimental neuroprotectors tested revealed to be more effective in the monoculture model compared to the co-culture system. To explain these differences, the authors formulated the following hypothesis: (i) some of the compounds used might be metabolized by astrocytes; (ii) the presence of astrocytes could affect the metabolic ability of neurons; (iii) astrocytes might also have contributed to alter the cell death mechanisms; (iv) the continuous conversion of MPTP into MPP+ by astrocytes could have contributed to an early stress response or adaptations [173]. Overall, the pharmacological experiments revealed that the MPTP co-culture model represents a valid model of degeneration of human dopaminergic neurons, capable of replicating crucial aspects of its toxicity in vivo [173]. Most importantly, the co-culture model was able to elucidate the pharmacological differences found between the models, due to their different complexities.

In 2021, Sergi and colleagues assessed the potential anti-apoptotic and anti-inflammatory role of the polyphenol trans-ε-viniferin (a dimer of resveratrol) in a co-culture model combining the rat pheochromocytoma neuronal cell line, PC12, and the microglial cell line N9. To mimic the PD phenotype they resorted to the compound 6-hydroxydopamine [175]. To study the impact of LPS-activated microglia on the survival rate of the neuronal cells, cells were co-cultured in a non-contact model. They showed that trans-ε-viniferin attenuated the activation of the apoptotic cascade induced by 6-hydroxydopamine in neurons and the neurotoxicity induced by LPS-activated microglia [175].

Johnson et al. performed an in vitro study to assess if others bioactive compounds of mucuna seeds (*Mucuna pruriens*), a well-established natural source of levodopa, may exert some biological effect in mitigating the symptoms of the disease [176]. For that, the authors prepared an extract of mucuna seeds containing a low concentration of levodopa and tested the effects on murine BV-2 microglia cells and SH-SY5Y neuroblastoma cells, individually and in a non-contact co-culture model [176]. In the BV-2 cells, in monoculture, the extract of mucuna seeds reduced the hydrogen peroxide-induced cytotoxicity, the formation of reactive oxygen species and the LPS-induced nitric oxide species release. In the neuroblastoma cells, the extract was able to reduce the 6-hydroxydopamine-induced oxidative stress [176]. In co-culture, exposure of LPS-activated BV-2 cells to the extract of mucuna seeds prevented apoptosis of neuroblastoma cells [176]. This suggests that mucuna seeds have other bioactive compounds, besides levodopa, with neuroprotective potential [176].

In 2016, Kuan and colleagues analyzed the influence of α-synuclein aggregation, a protein with innate neurotoxicity and a pathological hallmark in PD, on endothelial cells barrier function, using two co-culture models: (i) hCMEC/D3 cells co-cultured with primary rat astrocytes and (ii) hCMEC/D3 cells co-cultured with primary neuronal cultures [177]. Models were exposed to α-synuclein preformed fibrils (0,1 μg/μL, 14 days) to evaluate functional, molecular, and cellular changes. They showed a significant downregulation of ZO-1 and occludin expression by α-synuclein preformed fibrils [177]. Nonetheless, these changes in the protein expression did not significantly change TEER or permeability values. One explanation for this may lies in the compensation provided by claudin-5, another junctional protein that limits the paracellular crossing [177]. Thus, by using these models, the authors established a link between α-synuclein aggregation and the impaired inexpression of tight junction proteins, which may contribute to disease pathology [177].

In another study involving α-synuclein, Dohgu and colleagues, in 2019, compared an in vitro monoculture model of rat endothelial cells with a co-culture model of rat endothelial cells and rat pericytes, to study the influence of pericyte-derived inflammatory mediators on BBB disruption [178]. Exposure to α-synuclein (50 μg/mL, for 24 h) induced a considerable increase in endothelial permeability to sodium fluorescein in the co-culture model, only when added to the abluminal side containing pericytes [178]. In addition, α-synuclein exposure triggered IL-1β, IL-6, MCP-1, TNF-α, and MMP-9 release by pericytes, which caused the deregulation of the BBB barrier function. This study elucidated a role for pericytes, in response to α-synuclein, in inducing endothelial barrier dysfunction, via inflammatory mediators [178].

Matsumoto and colleagues, in 2012, used a co-culture model of primary rat endothelial cells and primary rat microglia cells to investigate the effect of LPS-activated microglia on P-gp expression and function [179]. A decline in P-gp activity is observed in patients with neurological diseases, such as Parkinson’s disease. In addition, microglia release pro-inflammatory mediators that interfere with BBB function, including the integrity of tight junction and efflux transporters’ activity [179]. The monoculture model of rat endothelial cells was compared to the co-culture model of rat endothelial cells with rat microglia in terms of rhodamine 123 intracellular accumulation, after microglia activation with LPS (1 ng/mL added to the abluminal side for 6 h) [179]. In the monoculture, no significant effects were detected on rhodamine 123 accumulation. Meanwhile, in the co-culture system, LPS substantially increased rhodamine 123 intracellular accumulation [179]. The addition of diphenyleneiodoniumchloride (DPI), an inhibitor of NADPH oxidase in microglia, reduced the accumulation of rhodamine 123 within the endothelial cells induced by LPS-activated microglia. This indicates that LPS-activated microglia cause the dysfunction of P-gp activity in endothelial cells, via NADPH oxidase, which may constitute a potential target in neuro-inflammatory diseases [179].

Liu and colleagues, in 2014, evaluated the limitation imposed by P-gp to the BBB crossing of the new drug candidate for PD treatment, FLZ (N-2-(4- hydroxy-phenyl)-ethyl]-2-(2,5-dimethoxy-phenyl)-3-(3-methoxy-4-hydroxy-phenyl)-acrylamide) [180]. They developed two in vitro models, designed to perform permeability studies in a normal and pathological BBB, by co-culturing: (i) normal physiological rat brain endothelial cells with C6 astrocytic cells and (ii) rat endothelial cells treated with 6-hydroxydopamine (2 μg/mL), with C6 astrocytic cells [180]. Co-culture of brain endothelial cells with astrocytes improved TEER values and the expression of efflux transporters, namely P-gp. The diffusion of FLZ (1, 5, 10 µM) across the normal and pathological model of the BBB occurred in both apical–basolateral and basolateral–apical directions. However, the basolateral–apical flux was considerably higher than that in the apical–basolateral direction, thus suggesting the presence of efflux pumps [180]. In addition, FLZ revealed a higher efflux ratio in the pathological model. P-gp inhibition with zosuquidar (5 µM) inhibited FLZ efflux, resulting in an intracellular accumulation of FLZ in rat endothelial cells [180]. In contrast, BCRP inhibition with FTC (10 µM) did not significantly change the intracellular accumulation of FLZ. These results indicated that the low permeability of the FLZ into the brain is due to P-gp efflux activity, limiting its brain intracellular accumulation. They also highlight that normal and 6-hydroxydopamine-treated rat brain endothelial cells represent good systems to study, in vitro, the BBB function under physiological and pathological conditions [180].

In recent years, the emergence of iPSC technology allowed several researchers to investigate PD using distinct iPSC models of the disease, particularly with human iPSC-derived neurons [181,182]. Apart from differentiating iPSCs into dopaminergic neurons, significant progress has also been achieved in the differentiation of iPSCs into other CNS resident cells, such as astrocytes and microglia-like cells, which facilitate disease modeling through heterotopic 2D cell–cell interaction models [182].

In the field of neurodegenerative diseases’ research, in particular in PD, several co-culture models have been developed using iPSC-derived cells, including co-cultures of iPSC-derived cortical neurons with iPSC-derived microglia [166], and co-cultures of human iPSC-derived astrocytes with microglia [183]. Indeed, and interestingly, Rostami and colleagues, in 2021, developed a co-culture model of human iPSC-derived astrocytes and microglia, which were exposed to preformed fibrils of α-synuclein (0.5 μM) or Aβ (0.2 μM), for 24 h, 4 days, or 7 days, and the results compared to those obtained for the monocultures of either cell type. When in monoculture, both microglia and astrocytes were capable of internalizing α-synuclein and Aβ aggregates, although microglia demonstrated to be more efficient in degrading the protein aggregates. When in co-culture, despite both cell types internalizing α-synuclein and Aβ fibrils, microglia exhibited a higher uptake of the aggregates. Outstandingly, when in co-culture, a decreased accumulation of intracellular deposits of α-synuclein and Aβ was depicted in astrocytes, when compared to the monocultures. Moreover, it was demonstrated that when in co-culture, astrocytes and microglia maintained continuous communication through tunneling nanotubes and other membrane structures. By live cell imaging, it was observed that microglia, upon adhering to the cell membrane of an astrocyte, could attract and eliminate intracellular protein deposits from the astrocyte. Therefore, this dynamic interaction between these two cell types promoted the transfer of protein aggregates from astrocytes to microglia, thus potentially representing a crucial mechanism for the clearance of protein aggregates within the affected brains. Overall, the collected data underscore a synergistic effect of astrocytes and microglia in processing Aβ and α-synuclein aggregates, emphasizing the significance of microglia and astrocyte interactions in the clearance of protein aggregates and, consequently, the importance of the glial cellular crosstalk in PD or even AD progression [183].

However, studies on the use of co-culture models for modelling PD using iPSC-derived BMECs are fairly scarce. For example, the previously described accelerated differentiation method developed by Hollman and colleagues, in 2017, for obtaining BMECs from iPSCs, was also found to ensure the long-term stability of BMECs containing biallelic PARK2 mutations linked to PD. Indeed, using the E6 medium, an iPSC line SM14, derived from a preclinical PD patient with compound heterozygous loss-of-function mutations in PARK2 linked to familial early-onset PD, was successfully differentiated into BMECs. However, although control iPSC-derived BMECs were tested in co-culture with both iPSC-derived astrocytes and primary human brain pericytes, the same was not performed for BMECs containing the biallelic PARK2 mutations [95].

Knowing that BBB dysfunction is clearly associated with several neurodegenerative diseases, particularly at late stages, and aiming at elucidating the impact of several mutations associated with familial neurodegenerative disorders on BBB function and their potential to induce BBB impairment, Katt and colleagues, in 2019, used iPSCs from three healthy individuals and eight individuals with neurodegenerative diseases [two from PD patients, two from ALS patients, two from AD patients, and two from Huntington’s disease (HD) patients) and differentiated them into BMECs [184]. The characteristics and barrier function of confluent monolayer human iPSC-derived BMECs were then assessed, namely TEER values and solute permeability (lucifer yellow, d-glucose, and rhodamine 123 permeabilities), the expression of BBB biomarkers at the mRNA and protein levels, as well as the P-gp efflux activity (rhodamine 123 efflux ratio and the effect of tariquidar-mediated P-gp inhibition). All iPSC-derived BMECs (obtained from healthy and disease iPSCs) expressed several BBB-associated markers, at both mRNA and protein levels, namely the tight junction proteins occludin, claudin-5, and ZO-1, as well as P-gp. TEER values obtained from healthy controls ranged between 1800 and 2500 Ω·cm^2^, while significantly decreased TEER values were obtained for the two ALS lines, and one of the PD, AD, and HD lines (TEER values ranging from 500 to 1000 Ω·cm^2^). Even though the TEER values obtained for the disease iPSCs-derived BMECs were fairly below the physiological values observed in animal models (1500–8000 Ω·cm^2^), they were relatively high when compared to those obtained for several primary and immortalized cells. The normal barrier function and restricted paracellular transport, evaluated through the assessment of lucifer yellow permeability, was confirmed for all the iPSC-derived BMECs, except for one AD (*PSEN1 A246E*) and one HD (*HTT CAG:50*) line, for which a small, but significant, decrease in barrier function was observed. The permeability of d-glucose, a measure of the GLUT1 activity, was significantly decreased in two of the tested disease iPSC-derived BMECs, and in one PD (*SCNA1* triplication) and another ASL (*C9orf72* expansion) model, when compared to heathy BMECs. Concerning P-gp, a reduced activity or incorrect polarization was suggested to occur for all disease lines, except SOD4AV (ALS), as evidenced by a rhodamine 123 efflux ratio smaller than 1, when compared to efflux ratios of 2–4 obtained for the three healthy control lines (the effectiveness of P-gp efflux activity is measured through the efflux ratio of the basolateral-to-apical and apical-to-basolateral permeabilities, and for P-gp substrates, the obtained value should be greater than 1.0). Upon tariquidar-mediated P-gp inhibition, a reduction in rhodamine 123 efflux ratio was observed for many of the tested iPSC-derived BMECs, although this effect was only significant for the healthy controls. Overall, the obtained data support the hypothesis that mutations associated with neurodegenerative diseases can independently induce BBB dysfunction, further suggesting that the accumulation of defects in BMECs may ultimately lead to BBB impairment [184].

Astrocytes are vital elements of the neurovascular unit, providing support for a proper BBB functioning, and their pathological transition to reactive states can have both protective and detrimental effects on BBB function [185]. In fact, although inflammation triggered by microglia is clearly linked to neurodegeneration, inflammation associated with reactive astrocytes may also be significant [185], as demonstrated by the neuroprotective effect of inhibiting inflammatory astrocyte conversion in mouse models of ALS [186] and PD [187]. With that in mind, and knowing that the inflammation of brain endothelial cells is observed in both aging and neurodegenerative conditions, Kim and colleagues developed, in 2022, an iPSC-derived BBB co-culture model (iPSC-derived BMEC-like cells co-cultured with iPSCs-derived astrocytes) aiming to investigate the impact of astrocytes on BBB function. It was demonstrated that tumor necrosis factor (TNF) induced the conversion of astrocytes into an inflammatory reactive state, which lead to BBB dysfunction, as clearly demonstrated through the loss of passive barrier function and by the upregulation of vascular cell adhesion molecule 1 (VCAM-1). Moreover, BBB dysfunction relied on the activation of STAT3, a signaling pathway that is well established to mediate astrocyte reactivity in animal models. The activation of STAT3 signaling also pointed as a central axis for a subpopulation of reactive astrocytes in CNS disorders. Furthermore, SERPINA3 [encoding alpha 1-antichymotrypsin (α1ACT)], which showed increased expression, was identified as a candidate astrocyte factor causing the observed BBB disruption [185]. Overall, the obtained data clearly demonstrated that the developed iPSC-derived cell co-culture model can be successfully applied for the in vitro evaluation of the impact of astrocytes in BBB disruption over the course of disease, namely the implication of the inflammation associated with reactive astrocytes in BBB dysfunction and neurodegeneration.

## 4. Concluding Remarks

The BBB is a complex cellular barrier formed by endothelial cells under the influence of different cell types that together give life to the so-called “neurovascular unit”. The BBB is responsible for protecting the brain from potentially toxic agents (both endobiotics and xenobiotics) and pathogens present in the blood circulation, while providing it with the optimal energetic and nutritional conditions for normal functioning. Its barrier status is due to several characteristics that make it unique, particularly the lipid nature of its membrane and the tight junction proteins that limit the permeability of hydrophilic compounds; the presence of SLC and ABC transporters that regulate the transcellular transport of nutrients, substrates and unwanted products, and also the presence of specific mechanisms that regulate the entry of required proteins and peptides; the presence of enzymes (e.g., CYP450) that give it additional metabolic protection and its ability to regulate the CNS immune defenses.

Since endothelial cells were first successfully isolated from brain microvessels, several useful models have been developed, although with limitations inherent to their origin, isolation, and immortalization process. Cell culture-based models have allowed the expansion of knowledge on the pathophysiology of the BBB, providing an excellent alternative to the limited means of in vivo studies and an excellent tool in neurotoxicity, neurodegeneration, and new CNS drug research. The development of more realistic in vitro models that can recapitulate the complexity and dynamics of the BBB has driven over the past few years’ changes in the field of BBB modeling, with the introduction of new technologies and methods to improve the existing models and to create new ones. The switch to multicellular models encompassing several cell types of the neurovascular unit, mainly astrocytes and pericytes, in culture with endothelial cells was an important breakthrough in bringing the characteristics of the models closer to those of the human BBB phenotype.

Given its complexity, this barrier stands as a real challenge to researchers and the fact that its true phenotype is far from being completely known, makes it difficult to create an ideal model. More recently, stem cell studies aimed at establishing a functional and reliable model, and 3D technology, which have been advancing rapidly and gaining some popularity by being used in BBB model engineering, mark the future perspectives.

These most recent efforts reinforce that, despite all the shortcomings encountered and the gaps that remain to be fulfilled about this dynamic barrier, the ongoing pursuit of the ideal in vitro model to expand the portfolio concerning the many aspects related to the BBB continues. The creation of in vitro models, using cell cultures that are increasingly reliable and realistic of their complex properties becomes imperative. This priority takes on special emphasis in studies related to the pathophysiology of neurodegenerative diseases and, nonetheless, in the development of new drugs that target the CNS.

Over the past few years, several studies have elucidated the importance of cell–cell communication for a better understanding of the pathogenesis of diseases, especially in the most prevalent ones, such as Alzheimer’s and Parkinson’s diseases. This may facilitate not only the discovery of new therapeutic targets, but also to generate better outcomes regarding toxicological assays.

## Figures and Tables

**Figure 1 biomedicines-12-00626-f001:**
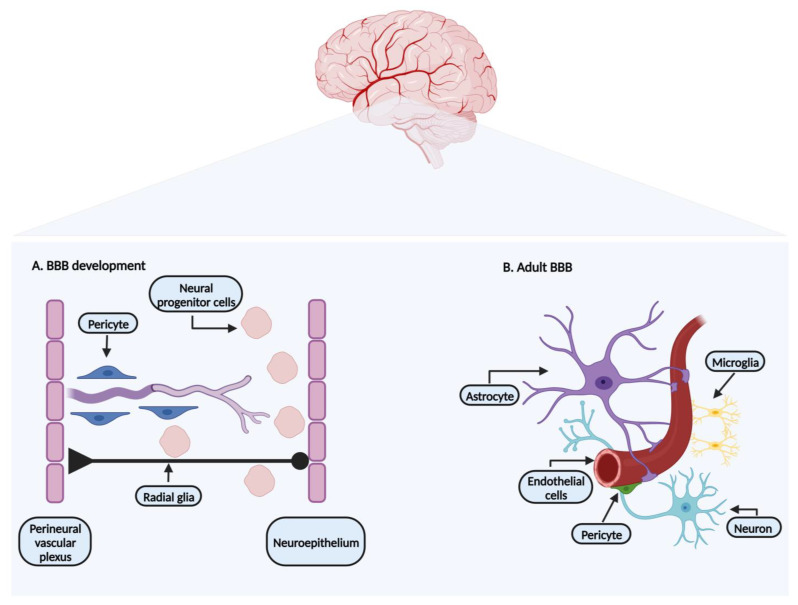
Schematic representation of the blood–brain barrier (BBB) development in the embryonic phase (**A**) and adult life (**B**). (**A**) The BBB development showing the interaction between immature endothelial cells, neural progenitors, pericytes, and radial glia. The angiogenesis from the perineural vascular plexus starts with migration towards the neuroepithelium. Newly forming blood vessels recruit pericytes for stabilization. In parallel, originating from neuroepithelium and using radial glial cells as a guidance structure, neural progenitor cells migrate and begin their differentiation. (**B**) Adult BBB representation showing a more elaborate structure. Endothelial cells, astrocytes, microglia, pericytes, and neurons constitute the neurovascular unit. Created with Biorender.com.

**Figure 2 biomedicines-12-00626-f002:**
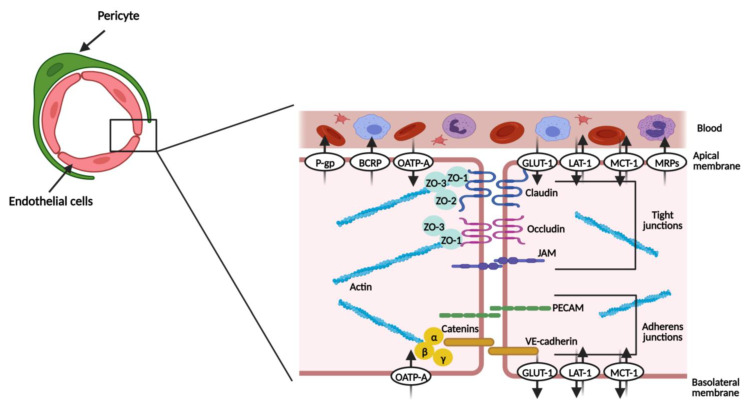
Schematic representation of the characteristics of the endothelial cells of the blood–brain barrier: tight junctions, adherens junctions, and influx and efflux transporters. Endothelial cells (surrounded by a pericyte) are connected by the inter-endothelial junctions. The tight junctions comprise the proteins claudin, occludin, and junctional adhesion molecule (JAM). Endothelial cells are also connected by catenins, vascular endothelial cadherin (VE-cadherin), and platelet endothelial cell adhesion molecules (PECAMs), which constitute the adherens junctions. The figure also illustrates efflux transporters P-glycoprotein (P-gp), breast cancer resistance protein (BCRP), and multidrug resistance-associated proteins (MRPs), as well as some of the main influx carriers such as the glucose transporter (GLUT-1), large amino acid transporter (LAT-1), monocarboxylate transporter (MCT-1), and organic anion transporting polypeptide (OATP-A). Created with Biorender.com.

**Figure 3 biomedicines-12-00626-f003:**
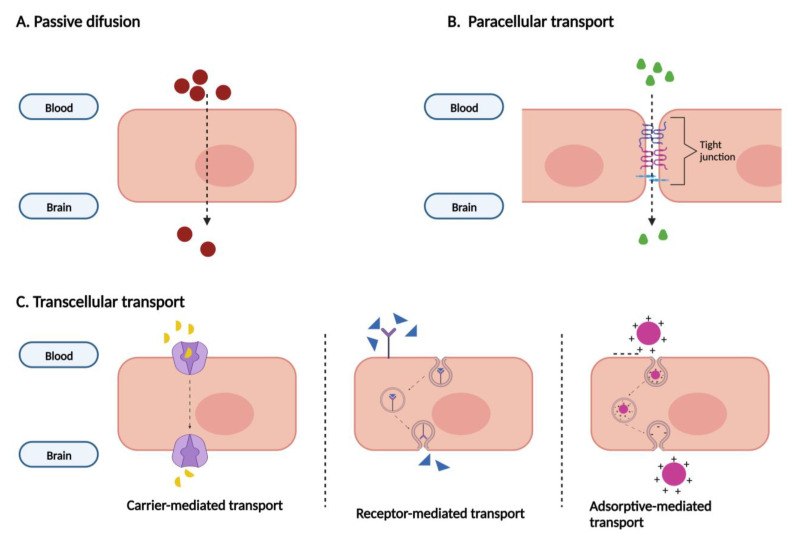
Illustration of the transport pathways at the blood–brain barrier. (**A**) Passive diffusion through which lipid-soluble compounds can cross, given the large surface area and the lipophilic nature of the membrane of the endothelium; (**B**) Paracellular transport, which allows water-soluble compounds to cross; (**C**) Transcellular transport that comprises: (1) carrier-mediated transport, including transport proteins for glucose, amino acids, purine bases, and other substances; (2) receptor-mediated transport, reserved for certain proteins as transferrin and insulin; (3) adsorptive-mediated transport, allowing to albumin and other plasma proteins enter the CNS. Created with Biorender.com.

**Figure 4 biomedicines-12-00626-f004:**
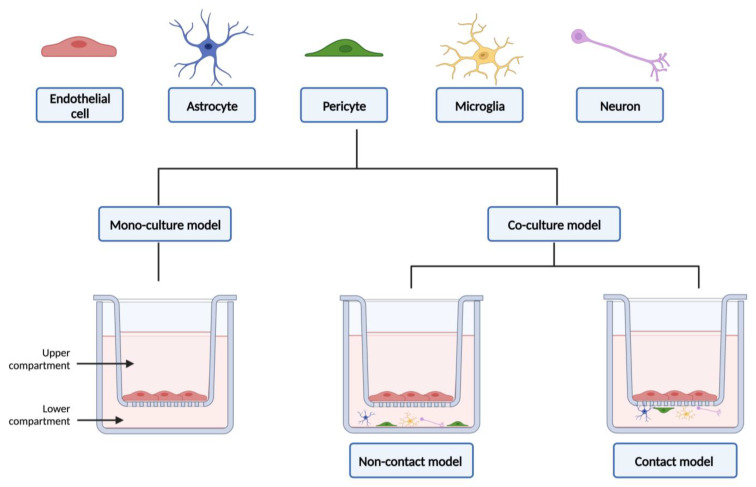
Representative scheme of the static model employing the Transwell systems. The Transwell inserts can be used for culturing brain endothelial cells, in monoculture models, for drug permeability analyses. The endothelial cells are seeded in the upper compartment, which represents the “blood side”, while the lower compartment is reserved to represent the “brain part”. On the other hand, the inserts can be used for co-culture models, either in contact or non-contact. The other cells that can be used (astrocytes, pericytes, microglia, and neurons) can improve the BBB phenotype. Created with Biorender.com.

**Table 1 biomedicines-12-00626-t001:** Characteristics and main functions of the principal proteins of the tight junctions (claudins, occludin, and junctional adhesion molecule) and adherens junctions (vascular endothelial cadherin, catenins, and platelet endothelial cell adhesion molecules) of the blood–brain barrier endothelial cells.

Inter-Endothelial Junctions	Principal Proteins	Characteristics/Main Functions	References
**Tight** **junctions**	**Claudins**	- Family of membrane proteins that include more than 20 proteins, although only claudin 1, 3, 5, and 12 seem to be important for BBB function; - Induce and seal the tight junctions; - Are responsible for limiting the paracellular movement.	[3,9,11,15]
	**Occludin**	- Controls calcium transport across the BBB; - Influences the paracellular permeability, maintaining stability and the barrier function.	[15,16]
	**JAM**	- Includes JAMs 1, 2, and 3 and regulate cell polarity; - Controls the transmigration of inflammatory cells, namely leukocytes, through the BBB.	[3,9,15]
**Adherens** **junctions**	**VE-cadherin**	- Also referred to as cadherin-5, is the main membrane protein of the adherens junctions; - Contributes to the stabilization of blood vessels and vascular permeability.	[3,9,15]
	**Catenins**	- Include α, β, and γ-catenin; - α-catenin regulates the binding functions of cadherins; - β-catenin helps to stabilize the adherent junctions, because it inhibits the proteolysis of cadherins.	[9,15]
	**PECAMs**	- Expressed in all endothelial cells; - Maintain the homeostasis of barrier function in the endothelial cells; - Help to restore the barrier integrity of BBB after perturbations.	[3,15]

JAM—junctional adhesion molecule; VE-cadherin—vascular endothelial cadherin; PECAMs—platelet endothelial cell adhesion molecules.

**Table 2 biomedicines-12-00626-t002:** Efflux transporters expressed in the blood–brain barrier and their main characteristics.

ABC Transporter	Main Functions/Characteristics	References
**P-gp** **(ABCB1)**	- Product of the *MDR1* gene highly expressed on the membrane of endothelial cells; - Wide-ranging substrate selectively, which allows the protection of the CNS against endogenous substrates and xenobiotics; - The classes of substrates include mostly amphipathic cations and organic molecules; - Important role in neuroprotection and resistance to therapeutic drugs.	[21,22,26]
**BCRP** **(ABCG2)**	- Responsible for the efflux of endogenous and exogenous compounds; - Substrate overlap with P-gp, limiting the BBB permeability (synergistic effect).	[12,21,22,26]
**MRPs** **(ABCC)**	- Family of several members, although the subclass of transporters noted for their relevance in drug delivery to the CNS are MRP1, 2, 3, 4, 5, and 6; - Each MRP possesses a specific substrate affinity; - The classes of substrates include unconjugated organic anions or their glutathione, glucuronide, sulfate, or phosphate conjugates; - MRP4 also transports amphiphilic anions (prostaglandins), and steroid and bile salts metabolites; - MRP4 and 5 may also transport nucleotides.	[12,21,22,26]

P-gp—P-glycoprotein; BCRP—breast cancer resistance protein; MRPs—multidrug resistance-associated proteins family.

**Table 3 biomedicines-12-00626-t003:** Solute-carrier (SLC) transporters expressed in the blood–brain barrier and their main characteristics.

ABC Transporter	Main Functions/Characteristics	References
**GLUT-1** **(SLC2A1)**	- Glucose transporter-1 (GLUT-1); - Sodium-independent transporter, which transports glucose through a mechanism of facilitated diffusion; - Responsible for providing a constant flux of glucose to the brain.	[3,27]
**SGLT1** **(SLC5A1)**	- Sodium-dependent glucose transporter-1 (SGLT1); - Sodium-dependent transporter that uses the energy from the sodium gradient created by the sodium/potassium ATPase to promote the co-transport of sodium and glucose into the cell; - Expressed under pathological conditions (e.g., ischemic conditions) or GLUT-1 mutations, promoting the glucose delivery to the brain.	[27]
**MCT1** **(SLC16A1)**	- Monocarboxylate transporter-1 (MCT1); - Its activity is pH-dependent (active at acidic pH), which supports a symport process with protons; - Responsible for the uptake of ketone bodies (another energetic substrate), lactate, pyruvate, β- hydroxybutyrate, and short-chain fatty acids as acetate; - It also removes lactate from the brain, helping to control the brain acidosis.	[3,27]
**CRT** **(SLC6A8)**	- Creatine transporter (CRT); - Creatine was described as an important compound for energy metabolism, via the creatine/phosphocreatine kinase system; - Recent studies described creatine as a potential neuromodulator and osmotic regulator in the brain.	[27,28]
**LAT-1** **(SLC7A5)**	- Large amino acid transporter-1 (LAT-1); - Transports amino acids and L-Dopa (1,3,4-dihydroxyphenylalanine). Its expression on the BBB is important in Parkinson’s disease.	[3,27]
**ATB ^0,+^** **(SLC6A14)**	- Labeled as amino acid transporter characterized by a large substrate specificity: neutral (“0”) and basic (“+”) amino acids (ATB ^0,+^); - Also transports carnitine, and acetylcarnitine (precursor of some neurotransmitters).	[27]
**CAT-1** **(SLC7A1)**	- Cationic amino acid transporter-1 (CAT-1); - Responsible for the transport of arginine, lysine, and histidine.	[27]
**EAAT** **(SLC1)**	- Excitatory amino acid transporter (EAAT); - Transports acidic amino acids as L-glutamate; - Plays an important role in protecting the brain against excitotoxicity by effluxing glutamate, thus helping to regulate its levels.	[3,27]
**TAUT** **(SLC6A6)**	- Taurine transporter (TAUT); - Recent studies highlight the role of taurine in osmoregulation.	[27]
**CTL1** **(SLC44A1)**	- Choline transporter-like protein (CTL1); - Allows the transport of choline to the brain, necessary for the synthesis of acetylcholine and choline-containing lipids, as phosphatidylcholine and sphingomyelin.	[3,27]
**OAT-3** **(SLC22A8)**	- Organic anions transporter-3 (OAT-3); - Responsible for the uptake of drugs and, additionally, responsible for the efflux of a uremic toxin (indoxyl sulfate) from the brain.	[27]
**OATP** **(SLC21/SLCO)**	- Organic anion transporting polypeptide (OATP); - Transports several endogenous and exogenous amphipathic acidic compounds to the brain; - OATP-A is responsible for the efflux of phase II metabolites, including glucuronides, sulfates, and glutathione conjugates.	[3,27]
**OCT1** **(SLC22A1)** **OCT2** **(SLC22A2)**	- Organic cation transporter (OCT); - Facilitate the influx of choline (also a substrate of these transporters).	[27]

**Table 4 biomedicines-12-00626-t004:** Markers expressed in cell-culture-based models of the blood–brain barrier.

Markers	Essential Expression in a BBB Model	Validation	Observations/Limitations	Relevance	References
**Tight** **junctions**	Occludin Claudin-5 ZO-1	Trans-endothelial electrical resistance (TEER)	- Not damaging to the cells, when measuring the ionic resistance of cells in monolayer; - Impedance is calculated and normalized to the surface area applying a voltage between electrodes placed in the apical and basolateral compartments in a classical Transwell assay; - May differ between studies due to differences in the junctional proteins expressed; differences in the measuring equipment (e.g., chopstick electrodes); temperature and handling of cells during analysis; - Difficulties in translating to a functional estimation of tightness, because the monolayer tightness itself, which depends on the composition of tight junctions and on the size of the compound under study, also interferes with the estimation	- Studies of tight junctions; - Transport and uptake studies.	[29,30]
		Permeability	- Defined as solute fluidity across the unit area under a unit concentration gradient; - Many primary and immortalized cell lines do not exhibit TEER values, so permeability contributes to both paracellular and transcellular assessment; - Evaluation by hydrophilic tracer molecules, such as lucifer yellow, sodium-fluorescein, sucrose, or mannitol; - Inversely correlated with TEER, depending on the size of the molecule and the experimental design.		
**ABC** **transporters**	P-gp BCRP MRPs	- mRNA and protein expression; - Permeability experiments with a substrate for the pump (confirmation of the efflux ratio by introducing an inhibitor).	- Specifically the P-gp pump demonstrates the potential for efflux evaluation; - Essential to regulate the permeability of small lipophilic molecules by passive diffusion; - Generally polarized on luminal surface and capable of expelling a wide range of compounds.	- Transport and uptake/efflux studies; - Drug delivery through the BBB; - Toxicity assessments.	[29,30]
**SLC** **transporters**	GLUT-1 LAT-1 MCT-1	- mRNA and protein expression; - Cellular uptake in the presence/absence of inhibitors.	- GLUT-1 confirmation is essential to facilitate the transport of glucose (metabolic activity).	- Transport and uptake studies; - Drug delivery through the BBB; - Nutrition studies.	[29,30]
**Receptor** **systems**	Transferrin receptor	- mRNA and protein expression; - Transferrin uptake—transport of iron.	- Highly expressed in brain endothelial cells in vivo; - Several receptor systems potentially capable to mediate transcytosis, including the insulin receptor, although the most studied is the transferrin receptor; - Studies confirmed that facilitates CNS delivery of compounds with clinical relevance.	- Studies of receptor-mediated transport; - Brain nutrition studies.	[29,30]

ZO—zonula occludens; mRNA—messenger ribonucleic acid; P-gp—P-glycoprotein; BCRP—breast cancer resistance protein; MRPs—multidrug resistance-associated proteins; GLUT—glucose transporter; LAT—large amino acid transporter; MCT—monocarboxylate transporter.

**Table 5 biomedicines-12-00626-t005:** Advantages and disadvantages for each cell-based model.

Model Type	Origin	Advantages	Disadvantages	References
**Primary cells**	Animal	- Generally characterized by high TEER and low paracellular permeability, when in a co-culture system; - High correlation with in vivo models.	- Time consuming and cost-intensive isolation process; - Interspecies variabilities, especially in protein expression and barrier properties; - High consumption of animals (rodents); - Finite lifespan.	[13,31,32,33,34]
	Human	- Same as above; - Expression of human-specific transporters, receptors, or immunological features.	- Ethical issues; - Difficult to acquire/obtain regularly; - Restricted availability of human material for the isolation of primary cells.	
**Immortalized** **cell lines**	Animal or Human	- Easily grown; - Usable over many passages; - Higher reproducibility of results, when compared with primary cells; - Better availability and scalability; - Commercially available.	- Lower values of TEER; - Incomplete tight junctions; - A tendency to lose their properties after a long period of passages; - Lower expression of some BBB transporters and enzymes, leading to a decreased generation of tight monolayers, and thus to a meaningful inadequate barrier function.	[13,31,32,33,34]

TEER—trans-endothelial electrical resistance; BBB—blood–brain barrier.

**Table 6 biomedicines-12-00626-t006:** Characteristics of the Bovine Brain Microvascular Endothelial Cells (BBMECs).

Category	Markers	References
**Enzymes**	- Alkaline phosphatase; - γ-glutamyl transpeptidase; - Catechol-O-methyl-transferase.	[36,37]
**Tight junctional proteins**	- ZO-1; - Occludin; - Claudin-1 and 5; - Low permeability to sucrose; - Tightness of the monoculture is variable (may limit their use in drug permeabilities studies).	[37,40,41]
**ABC transporters**	- P-gp; - MRP1, 4, 5 and 6; - BCRP.	[37,40,41]
**SLC transporters**	- GLUT-1; - SGLT1.	[37,40,41]

ZO—zonula occludens; P-gp—P-glycoprotein; MRP—multidrug resistance-associated protein; BCRP—breast cancer resistance protein; GLUT—glucose Transporter; SGLT—sodium-dependent glucose transporter.

**Table 7 biomedicines-12-00626-t007:** Characteristics of the Porcine Brain Microvascular Endothelial Cells (PBMECs).

Category	Markers	References
**Enzymes**	- Alkaline phosphatase; - γ-glutamyl transpeptidase.	[43,44,45,46]
**Tight junctional** **proteins**	- ZO-1; - Occludin; - Claudin-1 and 5; - High TEER values when compared with other primary cells and some immortalized cell lines; - Low permeability to sucrose.	[43,44,45,46]
**ABC transporters**	- P-gp; - BCRP; - MRP1, 2, and 4.	[43,44,45,46]
**SLC transporters**	- LAT-1 and 2; - GLUT-1; - OCT-1.	[43,44,45,46]

ZO—zonula occludens; P-gp—P-glycoprotein; BCRP—breast cancer resistance protein; MRP—multidrug resistance-associated protein; LAT—large amino acid transporter; GLUT—glucose transporter; OCT—organic cation transporter.

**Table 8 biomedicines-12-00626-t008:** Characteristics of the Rat Brain Microvascular Endothelial Cells (RBMECs).

Category	Markers	References
**Tight junctional** **proteins**	- ZO-1; - Low permeability to sucrose.	[47,48,49,50]
**ABC transporters**	- P-gp; - BCRP; - MRP1 and 4.	[47,48,49,50]
**SLC transporters**	- GLUT-1; - OATP-1 and 2; - OAT-1 and 3.	[47,48,49,50]

ZO—zonula occludens; P-gp—P-glycoprotein; BCRP—breast cancer resistance protein; MRP—multidrug resistance-associated protein; GLUT—glucose Transporter; OATP—organic anion transporting polypeptide; OAT—organic anion transporter.

**Table 9 biomedicines-12-00626-t009:** Characteristics of the Human Brain Microvascular Endothelial Cells (HBMECs).

Category	Markers	References
**Enzymes**	- Alkaline phosphatase; - γ-glutamyl transpeptidase; - Carbonic anhydrase IV.	[51,52,53]
**Tight junctional** **proteins**	- ZO-1 and 2; - Occludin; - Claudin-1, 3, and 5; - JAM-2; - High TEER values (when compared to other primaries cultures).	[51,52,53]
**ABC transporters**	- P-gp; - MRP1, 2, 4, 5, and 6.	[51,52,53]
**SLC transporters**	- LAT-1; - MCT-1; - GLUT-1.	[51,52,53]

ZO—zonula occludens; JAM—junctional adhesion molecule; P-gp—P-glycoprotein; MRP—multidrug resistance-associated protein; LAT—large amino acid transporter; MCT-1—monocarboxylate transporter-1; GLUT—glucose transporter.

**Table 10 biomedicines-12-00626-t010:** Characteristics of the BB19 immortalized cell line.

Category	Markers	References
**Tight junctional proteins**	- ZO-1.	[35,54]
**ABC transporters**	- P-gp; - MRP1, 2, 4 and 5; - BCRP.	[35,54]
**SLC transporters**	- OAT3 and 4.	[35,54]

ZO—zonula occludens; P-gp—P-glycoprotein; MRP—multidrug resistance-associated protein; BCRP—breast cancer resistance protein; OAT—organic anion transporter.

**Table 11 biomedicines-12-00626-t011:** Characteristics of the RBE4 immortalized cell line.

Category	Markers	References
**Enzymes**	- Alkaline phosphatase; - γ-glutamyl transpeptidase; - High levels of glutathione S-transferase.	[34,54,55,56]
**Tight junctional** **proteins**	- Occludin; - Claudin-2 and 15.	[34,54,55,56]
**ABC transporters**	- P-gp; - MRP1, 3, 4, and 5.	[34,54,55,56]
**SLC transporters**	- GLUT-1 and 3; - MCT1 and 6; - OAT1; - LAT1; - AUT; - CRT.	[34,54,55,56]
**Receptor-mediated transcytosis**	- Transferrin receptor.	[56,57]

P-gp—P-glycoprotein; MRP—multidrug resistance-associated protein; GLUT—glucose transporter; MCT—monocarboxylate transporter; OAT—organic anion transporter; LAT—large amino acid transporter; TAUT—taurine transporter; CRT—creatine transporter.

**Table 12 biomedicines-12-00626-t012:** Characteristics of the b.End3 immortalized cell line.

Category	Markers	References
**Tight junctional proteins**	- ZO-1 and 2; - Occludin; - Claudin-5; - JAM.	[57,58]
**ABC transporters**	- P-gp; - MRP1 and 5.	[57,58]
**SLC transporters**	- GLUT-1; - MCT1 and 2; - OAT1; - OATP1; - LAT1.	[57,58]

ZO—zonula occludens; JAM—junctional adhesion molecule; P-gp—P-glycoprotein; MRP—multidrug resistance-associated protein; GLUT—glucose transporter; MCT—monocarboxylate transporter; OAT—organic anion transporter OATP—organic anion transporting polypeptide; LAT—large amino acid transporter.

**Table 13 biomedicines-12-00626-t013:** Characteristics of the b.End5 immortalized cell line.

Category	Markers	References
**Tight junctional proteins**	- ZO-1; - Occludin; - Claudin-1.	[37,60]
**ABC transporters**	- P-gp.	[37,60]
**SLC transporters**	- GLUT-1.	[37,60]

ZO—zonula occludens; P-gp—P-glycoprotein; GLUT—glucose transporter.

**Table 14 biomedicines-12-00626-t014:** Characteristics of the cEND immortalized cell line.

Category	Markers	References
**Tight junctional proteins**	- Occludin; - Claudin-1,3, 5 and 12.	[61,62,63,64]
**SLC transporters**	- GLUT-1.	[61,62,63,64]

GLUT—glucose transporter.

**Table 15 biomedicines-12-00626-t015:** Characteristics of the cerebEND immortalized cell line.

Category	Markers	References
**Tight junctional proteins**	- Occludin; - Claudin-1, 3, 5, and 12.	[62,65]
**ABC transporters**	- P-gp; - MRP4; - BCRP.	[29]
**SLC transporters**	- GLUT-1.	[62,65]

P-gp—P-glycoprotein; MRP—multidrug resistance-associated protein; BCRP—breast cancer resistance protein; GLUT—glucose transporter.

**Table 16 biomedicines-12-00626-t016:** Characteristics of the hCMEC/D3 immortalized cell line.

Category	Markers	References
**Tight junctional** **proteins**	- Claudin-3 and 5; - JAM-1; - ZO-1 and 2; - Occludin.	[66,67,69]
**ABC transporters**	- P-gp; - MRP1, 4, and 5; - BCRP.	[66,67,69]
**SLC transporters**	- GLUT-1; - OCT1, 2, and 3; - ATB ^0,+^; - MCT1 and 3.	[66,67,69]
**Receptor-mediated transcytosis**	- Transferrin receptor.	[66,67,69]

JAM—junctional adhesion molecule; ZO—zonula occludens; P-gp—P-glycoprotein; MRP—multidrug resistance-associated protein; BCRP—breast cancer resistance protein; GLUT—glucose transporter; OCT—organic cation transporter; ATB—amino acid transporter; MCT—monocarboxylate transporter.

**Table 17 biomedicines-12-00626-t017:** Characteristics of the TY08 immortalized cell line.

Category	Markers	References
**Tight junctional proteins**	- Occludin; - Claudin-5; - ZO-1 and 2; - Low permeability to inulin.	[72,73]
**ABC transporters**	- P-gp; - BCRP; - MRP1, 2, 4, and 5.	[72,73]
**SLC transporters**	- GLUT-1; - CAT-1; - LAT-1; - MCT-1; - CRT.	[72,73]

ZO—zonula occludens; P-gp—P-glycoprotein; BCRP—breast cancer resistance protein; MRP—multidrug resistance-associated protein; GLUT—glucose transporter; CAT—cationic amino acid transporter; LAT—large amino acid transporter; MCT—monocarboxylate transporter; CRT—creatine transporter.

**Table 18 biomedicines-12-00626-t018:** Characteristics of the TY09 immortalized cell line.

Category	Markers	References
**Tight junctional** **proteins**	- Occludin; - Claudin-5 and 12; - ZO-1 and 2 (mRNA expressed at very low level); - restricted paracellular barrier properties.	[73]
**ABC transporters**	- P-gp; - BCRP; - MRP1, 4, and 5.	[73]
**SLC transporters**	- GLUT-1; - CAT-1; - LAT-1; - MCT-1; - CRT.	[73]

ZO—zonula occludens; mRNA—messenger ribonucleic acid; P-gp—P-glycoprotein; BCRP—breast cancer resistance protein; MRP—multidrug resistance-associated protein; GLUT—glucose transporter; CAT—cationic amino acid transporter; LAT—large amino acid transporter; MCT—monocarboxylate transporter; CRT—creatine transporter.

**Table 19 biomedicines-12-00626-t019:** Characteristics of the TY10 immortalized cell line.

Category	Markers	References
**Tight junctional proteins**	- Occludin; - Claudin-5; - ZO-1; - JAM-1; - Moderately restrictive barrier.	[35,77,78]
**ABC transporters**	- P-gp; - BCRP; - MRP1.	[35,77,78]
**SLC transporters**	- GLUT-1.	[35,77,78]

ZO—zonula occludens; JAM—junctional adhesion molecule; P-gp—P-glycoprotein; BCRP—breast cancer resistance protein; MRP—multidrug resistance-associated protein; GLUT—glucose transporter.

**Table 20 biomedicines-12-00626-t020:** Characteristics of the HBEC-5i immortalized cell line.

Category	Markers	References
**Tight junctional proteins**	- Occludin; - Claudin-5; - ZO-1;	[80,81,82]
**ABC transporters**	- P-gp; - BCRP; - MRP1 and 2.	[80,81,82]

ZO—zonula occludens; P-gp—P-glycoprotein; BCRP—breast cancer resistance protein; MRP—multidrug resistance-associated protein.

**Table 21 biomedicines-12-00626-t021:** Characteristics of the HBMEC/cibeta immortalized cell line.

Category	Markers	References
**Tight junctional proteins**	- Claudin-3, 5 and 12; - Occludin; - ZO-1; - JAM-1.	[66,83,84]
**ABC transporters**	- P-gp; - BCRP.	[66,83,84]
**SLC transporters**	- GLUT-1; - LAT-1; - OATP1 and 2.	[66,83,84]
**Receptor-mediated transcytosis**	- Transferrin receptor; - Insulin receptor.	[66,83,84]

ZO—zonula occludens; JAM—junctional adhesion molecule; P-gp—P-glycoprotein; BCRP—breast cancer resistance protein; GLUT—glucose transporter; LAT—large amino acid transporter; OATP—organic anion transporting polypeptide.

**Table 22 biomedicines-12-00626-t022:** Characteristics of the HBMEC/ci18 immortalized cell line.

Category	Markers	References
**Tight junctional proteins**	- Claudin-5; - Occludin; - ZO-1.	[85,86]
**ABC transporters**	- P-gp; - BCRP; - MRP4.	[85,86]
**SLC transporters**	- GLUT-1; - MCT-8.	[85,86]
**Receptor-mediated transcytosis**	- Transferrin receptor; - Insulin receptor.	[85,86]

ZO—zonula occludens; P-gp—P-glycoprotein; BCRP—breast cancer resistance protein; MRP—multidrug resistance-associated protein; GLUT—glucose transporter; MCT—monocarboxylate transporter.

**Table 23 biomedicines-12-00626-t023:** Characteristics of the SH-SY5Y immortalized cell line.

Phenotype	Markers	References
**Dopaminergic**	- Tyrosine hydroxylase; - Dopamine-β-hydroxylase; - Dopamine transporter; - Dopamine receptor 2 and 3; - Phenotype induced by treatment with retinoic acid co-administrated with other agents, like phorbol esters.	[88,89,90]
**Cholinergic**	- Expression of both muscarinic and nicotinic receptors; - Choline acetyltransferase; - Acetylcholine transporter; - Acetylcholinesterase; - Vesicular monoamine transporter; - G-protein coupled muscarinic receptor; - Ion channel nicotinic acetylcholine receptor; - Phenotype induced by treatment with retinoic acid, brain-derived neurotrophic factor (BDNF) and phorbol esters.	[88,89]
**Adrenergic**	- Synthesis of noradrenaline; - Noradrenaline transporter; - Vesicular monoamine transporter; - Expression of tyrosine hydroxylase, which is essential, since it is responsible for the catalysis of dopamine, giving rise to noradrenaline and adrenaline; - Phenotype induced by treatment with retinoic acid, phorbol esters, and dibutyryl cyclic adenosine monophosphate (AMP).	[88,89]

**Table 24 biomedicines-12-00626-t024:** Characteristics of the Caco-2 immortalized cell line.

Category	Markers	References
**Enzymes**	- CYP450-related enzymes; - Glutathione-S-transferase; - Sulfotransferase 1A1; - UDP-glucuronosyltransferase; - Alkaline phosphatase.	[55]
**Tight junctional proteins**	- Occludin; - Claudin-1, 2, 3, 4, 7, 11, 15, and 16.	[55]
**ABC transporters**	- P-gp; - BCRP; - MRP1, 2, 3, 4, 5, and 6.	[55]
**SLC transporters**	- GLUT-1; - MCT-1; - CAT-1; - LAT-1; - TAUT; - CRT; - EAAT-1 and 3.	[55]

CYP450—cytochrome P450; UDP—Uridine 5′-diphospho; P-gp—P-glycoprotein; BCRP—breast cancer resistance protein; MRP—multidrug resistance-associated protein; GLUT—glucose transporter; MCT—monocarboxylate transporter; CAT—cationic amino acid transporter; LAT—large amino acid transporter; TAUT—taurine transporter; CRT—creatine transporter; EAAT—excitatory amino acid transporter.

**Table 25 biomedicines-12-00626-t025:** Characteristics of the Madin–Darby Canine Kidney (MDCK) immortalized cell line.

Category	Markers	References
**Enzymes**	- Low expression of CYP450-related enzymes; - Sulfotransferase 1A1; - Alkaline phosphatase; - γ-glutamyl transpeptidase.	[55,91]
**Tight junctional proteins**	- Occludin; - Claudin-1, 2, 3, 4, 5, 7, 15, 16, and 19; - ZO-1.	[55,91]
**ABC transporters**	- P-gp; - MRP1, 2, 3, 4, and 5.	[55,91]
**SLC transporters**	- GLUT-1; - CAT-1; - LAT-1; - TAUT; - CRT; - EAAT-3.	[55,91]

CYP450—cytochrome P450; ZO—zonula occludens; P-gp—P-glycoprotein; MRP—multidrug resistance-associated protein; GLUT—glucose transporter; CAT—cationic amino acid transporter; LAT—large amino acid transporter; TAUT—taurine transporter; CRT—creatine transporter; EAAT—excitatory amino acid transporter.

**Table 27 biomedicines-12-00626-t027:** The neurovascular unit cells most used in co-culture.

Type of the Neurovascular Unit Cells Used in Co-Culture	Advantage(s)	References
**Astrocytes**	- Significant improvements in TEER values; - Allow to obtain lower permeability values; - Able to induce the formation of tight junctions; - Upregulate the expression of brain endothelial enzymes (especially the γ-glutamyl transpeptidase); - Increase the expression and the polarized location of BBB transporters, particularly P-gp and GLUT-1; - Allow for more specialized studies, since they increase the endothelial expression of receptor-mediated transcytosis.	[29,32,88,100,101,102]
**Pericytes**	- Significant improvements in TEER values; - Induce the expression of the MRP6 transporter; - Induce the endothelial maturation; - Induce the secretion of MMPs in the endothelium.	[29,32,100]
**Neurons**	- Induce the production of BBB-related enzymes; - Influence the barrier properties, as they secrete some vasoactive substances, including VEGF.	[32,51,100]
**Microglia**	- Releasing pro-inflammatory mediators, they increase the barrier permeability and reduce the levels of some tight junctions, thus playing a key role in the study of the BBB breakdown in pathological states; - Being responsible for the secretion of pro-inflammatory and oxidative factors in response to several different stimuli and toxicants, their use in co-culture becomes relevant in studies addressing their role in neurotoxicity.	[51,103,104]

TEER—trans-endothelial electrical resistance; P-gp—P-glycoprotein; GLUT—glucose transporter; MRP—multidrug resistance-associated protein; MMPs—matrix metalloproteinases; VEGF—vascular endothelial growth factor.

**Table 28 biomedicines-12-00626-t028:** Main advantages and disadvantages of dynamic models of the blood–brain barrier: Dynamic in vitro (DIV) system; hanging-drop method; forced-floating method; matrices and scaffolds; agitation-based approaches and microfluidic technology “organ-on-a-chip”.

Types of Dynamic Models	Advantages	Disadvantages	References
**Dynamic in vitro (DIV) system**	- Low permeability to intraluminal polar compounds; - Allows to obtain higher TEER values; - Trivial extravasation of proteins; - Expression of functional and more specialized ion channels and efflux transporters.	- Its structure does not allow to performed drug permeability studies; - Time-consuming and requires technical skills; - Requires the use of a high number of cells; - By not allowing the view of the intraluminal compartment, does not enable the analysis of morphological and/or phenotypic changes of the endothelial cells.	[94,104]
**Hanging-drop method**	- Simple and easy to use; - Maximal reproducibility for producing spheroids per drop.	- Larger volumes (>50 µL) cannot be used, because the surface tension does not have enough capacity to hold the droplet.	[106]
**Forced-floating method**	- Simple method to obtain consistent spheroids; - Easily reproducible; - Precoated plates are available; - By manipulating the quantity of seeded cells, spheroids can have different sizes.	- Time-consuming process when there is a need for coating the plates; - The coating polymer needs to be dissolved and autoclaved prior to use; - The precoated plates are expensive, thus increasing the cost of spheroids production.	[106]
**Matrices and scaffolds**	- Relatively easy method; - The use of hydrogels offers a porous structure, allowing the nutrients, oxygen, and drugs (necessary to survival) to reach the cells and the removal of waste products; - Provides cell culture conditions to a better migration and organization of the cells.	- Non-uniformity of the spheroids; - Expensive for large-scale production of spheroids; - Batch-to-batch variability.	[106]
**Agitation-based** **approaches**	- Spinner flask bioreactor: medium can be changed, ensuring long-term survival of cells; the motion gives nutrients to the cells and allows the removal of waste products. - Rotating cell culture bioreactors: low sheer forces; simplicity and easiness to handling; large-scale production of spheroids.	- Spinner flask bioreactor: altered cell morphology due to the sheer force of the stirring; larger consumption of culture medium; non-consistency in the sizes of spheroids. - Rotating cell culture bioreactors: variability in the size of spheroids.	[106]
**Microfluidic technology “organ-on-a-chip”**	- Applicable in drug development and permeability assays, toxicity studies, cell culture, genetic assays, protein studies, intracellular signaling, and medical research of stem cells and tissue engineering; - The materials used are permeable to oxygen and nutrients, which enables better proliferation of cells.	- More expensive and complex technology to establish culture models; - Requires further optimization.	[106,107,108,109]

**Table 29 biomedicines-12-00626-t029:** Advantages of co-culturing different cell types with endothelial cells obtained from primary sources.

Source of Endothelial Cells	Co-Cultivated with	Advantages/Observations	References
**Bovine**	- Rat astrocytic C6 glioma cell line - Primary rat astrocytes - Rat primary cultures of mixed glial cells	- Astrocytes show to be important for the maintenance of cell-cell adhesion; - High junctional tightness and low paracellular values, when compared to the monoculture; - Expression of enzymatic activity (monoamine oxidase and γ-glutamyl transpeptidase); - Functional polarization of transporters, namely P-gp and receptor-mediated transcytosis; - In a study conducted with mixed glial cells, the model produced a strong correlation with in vivo permeability, exhibiting complex tight junctions, low permeability to sucrose, and expression of metabolizing enzymes, transporters, and active efflux pumps.	[32,39,41,110,111]
**Rat**	- Rat pericytes and astrocytes	- High TEER values and lower permeability, producing high levels of mRNA for key tight junction proteins (claudin-5 and occludin); - Expression of different transporters (e.g., P-gp and GLUT-1); - Allows to perform studies of toxicity, permeability, and interactions with efflux transporters.	[32,55]
**Porcine**	- Primary rat astrocytes - Rat astrocytes cells lines (CTX-TNA 2, C6 glioma) - Primary rat astrocytes and pericytes - Primary porcine astrocytes and pericytes	- Increased TEER values (higher with astrocytic cell lines); - When used in triple-culture, the TEER achieved the highest values; - Low permeability to sucrose; - Switch to spindle-like morphology; - Up-regulation of γ-glutamyl transpeptidase and alkaline phosphatase expression; - Via ultrastructural analyses, by transmission electron microscopy, it was possible to analyze intact intercellular tight junction complexes at cell–cell interfaces, identifying a more highly restrictive in vitro model by resorting to astrocyte cell line CTX-TNA 2, when compared with the use of C6 cell line; - High expression of receptor-mediated transcytosis in co-culture; - Astrocytes and pericytes from rat and porcine are equally good to induce the improvements referred to so far, but syngeneic models are preferable to eliminate possible differences between species.	[32,42,43,44,111,112,113,114]
	- SH-SY5Y neuroblastoma cell line	- Result in high TEER values and lower permeability coefficients of low molecular weight compounds, possibly by influencing tight junctions’ complexes.	
	- SH-SY5Y neuroblastoma cell line and pericytes	- Improved in vitro model of the BBB, showing enhanced barrier properties due to the secreted factors released as a consequence of the cell–cell interaction, leading to the identification of several angiogenic, angiostatic and neurotrophic factors; - No direct effect of pericytes on the higher expression of tight junctions; - The secretion of growing factors under co-culture with pericytes.	
**Human**	- Primary human astrocytes - Primary human astrocytes and pericytes - Primary human astrocytes, pericytes and neurons	- Significant increase in barrier tightness, given by an enhancement in the TEER values (the most significant elevation was achieved with the model containing astrocytes, pericytes and neurons); - Reduced permeability coefficients; - Higher expression of functional P-gp; - The model containing human astrocytes in co-culture has proven to allow studies of monocyte transmigration in the CNS.	[31,51,115,116,117]

P-gp—P-glycoprotein; mRNA—messenger ribonucleic acid; TEER—trans-endothelial electrical resistance; GLUT—glucose transporter.

**Table 30 biomedicines-12-00626-t030:** Advantages of co-culturing immortalized cell lines with other cell types.

Cell Line	Co-Cultivated with	Advantages/Observations	References
**RBE4**	- Primary rat astrocytes - Rat astrocytic C6 glioma cell line	- Increased expression of γ-glutamyl transpeptidase and alkaline phosphatase; - Lower sucrose permeability coefficient.	[32,56]
	- Primary rat neurons - Primary rat neurons and astrocytes	- Co-culture with neurons allows an induction, in endothelial cells, of occludin synthesis and sorting at cell periphery; - The inclusion of astrocytes induced a significant decrease in sucrose permeability.	
**HCMEC/D3**	- Human astrocytes - Human cerebral astrocyte cell line SC1810 - Immortalized human astrocyte cell line SVG-A - Immortalized human astrocyte cell line SVGmm	- Yielded low TEER values in co-culture with astrocytes, showing no additional benefit regarding TEER; - Improvement of the barrier properties by increasing the TEER values only achieved with the SC1810 cell line; - No significant improvements on vectorial transport of P-gp substrates.	[35,42,46,68,70,118,119,120,121]
	- Immortalized human pericyte cell line HBPCT	- No increase the tight junction resistance; - The cell line was unable to respond positively to stimuli from immortalized pericytic cells.	
	- Human medulloblastoma cell line VC312R - Human medulloblastoma cell line DAOY	- The presence of tumor cells influenced the development and the tightness of the barrier, but the effects were dependent on the type of cell line used; - The permeability of the BBB sightly increased only in the presence of DAOY cells; - TEER values obtained in both models were low; - Expression of transferrin receptor when co-cultured with the DAOY cell line.	
	- SH-SY5Y neuroblastoma cell line	- Significant decrease in the viability of endothelial cells (approximately 89%), although there was no activation of the initiator caspases 3 and 7.	
	- Human astrocytoma cell line U87	- Increase in the paracellular permeability; - Co-culture had no effects on the inflammatory chemokine profile; - Decrease inTEER values, which may indicate that the barrier function is affected by an impact on cell morphology and tight junction expression.	
	- Primary human peripheral blood mononuclear cells	- Used for evaluation of BBB functions, at both the endothelial and migrating cell level.	
**TY10**	- Immortalized human pericyte cell line HBPCT	- No changes in TEER values; - The cell line did not respond positively to the addition of another cell type.	[35,122]
	- Immortalized human astrocyte cell line SVG-A	- The same response was verified to the astrocyte cell line.	
**BB19**	- Immortalized human pericyte cell line HBPCT	- No increase in TEER values.	[35,122]
	- Immortalized human astrocyte cell line SVG-A	- No improvement in TEER values was observed, which may indicate that the cells do not respond favorably to the presence of another cell line.	
**b.END3**	- Rat astrocytic C6 glioma cell line	- Exhibit low TEER values; - Compromised integrity of cell layers, possibly by secretion of cytokines like TNF-α, which promotes the opening of the BBB.	[32,93,123]
**CerebEND**	- Rat astrocytic C6 glioma cell line	- Slight increase in TEER values.	[29]

P-gp—P-glycoprotein; TEER—transendothelial electrical resistance; TNF-α—tumor necrosis factor-alpha.

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
