# Peer review of "Co-Culture Models: Key Players in In Vitro Neurotoxicity, Neurodegeneration and BBB Modeling Studies"

_biomedicines, 2024, doi:10.3390/biomedicines12030626_

Round 1

Reviewer 1 Report

Comments and Suggestions for Authors

After discussing the BBB development and characterization in detail, this review provides an effective and detailed discussion of cell culture as an in vitro BBB model, as well as co-culture models as multicellular models in which multiple cell types are introduced.

By using the BBB co-culture models, the focus on BBB permeability evaluation, neurotoxicity assessments and research on neurodegenerative diseases is very interesting and worthwhile. The review data are convincing and interesting, and well discussed.

This review is so well written that it seems possible to publish it in its current form.

Author Response

Reviewer comments:

Reviewer #1:

After discussing the BBB development and characterization in detail, this review provides an effective and detailed discussion of cell culture as an in vitro BBB model, as well as co-culture models as multicellular models in which multiple cell types are introduced.

By using the BBB co-culture models, the focus on BBB permeability evaluation, neurotoxicity assessments and research on neurodegenerative diseases is very interesting and worthwhile. The review data are convincing and interesting, and well discussed.

This review is so well written that it seems possible to publish it in its current form.

Response to Reviewer: We acknowledge the Reviewer for the thoughtful and insightful feedback on our manuscript. We truly appreciate the time and effort in reviewing our work and acknowledge the valuable feedback.

Reviewer 2 Report

Comments and Suggestions for Authors

The review titled, " Co-culture Models: Key Players in In Vitro Neurotoxicity, Neurodegeneration and BBB modeling studies" by Monteiro et. al. reviews the extensive list of BBB models including barrier-forming endothelial cells and their co-culture with astrocytes, neurons, and pericytes.  The authors do a thorough job of reviewing several hallmark barrier models along with their applications in neurotoxicity assays and neurodegenerative diseases.  The review is well written overall but needs some moderate editing of the English language with several "awkward" sentences.  Furthermore, I have several comments/suggestions that could improve the review below:

1) Please add functional assays/readouts (ie TEER) to the endothelial tables (Table 6-25).

2) Add gray horizontal sections in each table linking reference to cell line to marker to readout.  This will make the tables much easier to read and directly link the information to the appropriate reference.

3) iPSC models of the BBB make up a significant portion to the field.  They are mentioned in the co-culture section, but they should be highlighted in the BMEC portion as well.

4)In Section 3.4, the text would be much improved if it was broken down into subsections (ie passive vs. active barrier) or subsection (Tight junctions, efflux, etc.)

5) Section 3.5 would be improved if iPSC-BBB models were added to the AD, PD, and ALS sections.

6) A page of all the abbreviations could be useful and thus eliminating the abbreviations listed after each table.

Thank you.

Comments on the Quality of English Language

There wasn't a significant amount of spelling errors, etc. but throughout the review there were multiple awkwardly worded sentences.

Author Response

Reviewer #2:

The review titled, " Co-culture Models: Key Players in In Vitro Neurotoxicity, Neurodegeneration and BBB modeling studies" by Monteiro et. al. reviews the extensive list of BBB models including barrier-forming endothelial cells and their co-culture with astrocytes, neurons, and pericytes.  The authors do a thorough job of reviewing several hallmark barrier models along with their applications in neurotoxicity assays and neurodegenerative diseases.  The review is well written overall but needs some moderate editing of the English language with several "awkward" sentences. Furthermore, I have several comments/suggestions that could improve the review below:

1) Please add functional assays/readouts (ie TEER) to the endothelial tables (Table 6-25)

Response to Reviewer: We respectfully acknowledge the suggestion of the Reviewer to add functional assays/readouts such as TEER to the endothelial tables (Table 6-25). However, we believe that the focus of our review is on the models themselves (specific characteristics for in vitro modeling of the BBB) and their applications, rather than specific functional assays, which could increase the complexity of the tables.

2) Add gray horizontal sections in each table linking reference to cell line to marker to readout.  This will make the tables much easier to read and directly link the information to the appropriate reference.

Response to Reviewer: We greatly acknowledge the Reviewer’s suggestion, and the tables were accordingly improved in the revised version of the manuscript.

3) iPSC models of the BBB make up a significant portion to the field. They are mentioned in the co-culture section, but they should be highlighted in the BMEC portion as well.

Response to Reviewer: We greatly acknowledge the Reviewer’s suggestion. Although iPSC models are mentioned in section “2.5. Future Perspectives: Microfluidic and Stem Cells” and in Table 26, a brief introduction on the potential of iPSC-derived BBB models was added to the revised version of the manuscript.

4) In Section 3.4, the text would be much improved if it was broken down into subsections (ie passive vs. active barrier) or subsection (Tight junctions, efflux, etc.).

Response to Reviewer: We greatly acknowledge the Reviewer’s suggestion. However, we consider that the introduction of different subsections is difficult to make, as this topic is related to neurotoxicity assessment in general, rather than passive vs. active barrier or to specific analyses.

5) Section 3.5 would be improved if iPSC-BBB models were added to the AD, PD, and ALS sections.

Response to Reviewer: We greatly acknowledge the Reviewer’s suggestion. Section 3.5 was improved by adding studies reporting the use of iPSC-BBB models for Alzheimer’s and Parkinson diseases, and Amyotrophic Lateral Sclerosis.

6) A page of all the abbreviations could be useful and thus eliminating the abbreviations listed after each table.

Response to Reviewer: We greatly acknowledge the Reviewer’s suggestion. However, we consider that the abbreviations listed at the end of each table is a requirement from the editorial office.